# Contrastive Laplacian Eigenmaps

**Hao Zhu**[†,§]   **Ke Sun**[§,†]   **Piotr Koniusz** [*,§,†]
[§]Data61/CSIRO   [†]Australian National University
allenhaozhu@gmail.com, sunk@ieee.org, piotr.koniusz@data61.csiro.au

## Abstract

Graph contrastive learning attracts/disperses node representations for similar/dissimilar node pairs under some notion of similarity. It may be combined with a low-dimensional embedding of nodes to preserve intrinsic and structural properties of a graph. In this paper, we extend the celebrated Laplacian Eigenmaps with contrastive learning, and call them COntrastive Laplacian EigenmapS (COLES). Starting from a GAN-inspired contrastive formulation, we show that the Jensen-Shannon divergence underlying many contrastive graph embedding models fails under disjoint positive and negative distributions, which may naturally emerge during sampling in the contrastive setting. In contrast, we demonstrate analytically that COLES essentially minimizes a surrogate of Wasserstein distance, which is known to cope well under disjoint distributions. Moreover, we show that the loss of COLES belongs to the family of so-called block-contrastive losses, previously shown to be superior compared to pair-wise losses typically used by contrastive methods. We show on popular benchmarks/backbones that COLES offers favourable accuracy/scalability compared to DeepWalk, GCN, Graph2Gauss, DGI and GRACE baselines.

## 1 Introduction

Celebrated graph embedding methods, including Laplacian Eigenmaps [5] and IsoMap [42], reduce the dimensionality of the data by assuming that it lies on a low-dimensional manifold. The objective functions used in studies [5, 42] model the pairwise node similarity [7] by encouraging the embeddings of nodes to lie close in the embedding space if the nodes are closely related. In other words, such penalties do not guarantee that unrelated graph nodes are separated from each other in the embedding space. For instance, Elastic Embedding [8] uses data-driven affinities for the so-called local distance term and the data-independent repulsion term.

In contrast, modern graph embedding models, often unified under the Sampled Noise Contrastive Estimation (SampledNCE) framework [33, 28] and extended to graph learning [41, 15, 50], enjoy contrastive objectives. By maximizing the mutual information between patch representations and high-level summaries of the graph, Deep Graph Infomax (DGI) [43] is a contrastive method. GraphSAGE [15] minimizes/maximizes distances between so-called positive/negative pairs, respectively. It relies on the inner product passed through the sigmoid non-linearity, which we argue below as suboptimal.

Thus, we propose a new **COntrastive Laplacian EigenmapS (COLES)** framework for unsupervised network embedding. COLES, derived from SampledNCE framework [33, 28], realizes the negative sampling strategy for Laplacian Eigenmaps. Our general objective is given as:

$$\mathbf{\Theta}^* = \arg\max_{\mathbf{\Theta}} \mathrm{Tr}(f_{\mathbf{\Theta}}(\mathbf{X})^\top \Delta\mathbf{W} f_{\mathbf{\Theta}}(\mathbf{X})) + \beta\Omega(f_{\mathbf{\Theta}}(\mathbf{X})). \tag{1}$$

$\mathbf{X} \in \mathbb{R}^{n \times d}$ in Eq. (1) is the node feature matrix with $d$ feature dimensions given $n$ nodes, $f_{\mathbf{\Theta}}(\mathbf{X}) \in \mathbb{R}^{n \times d'}$ is an output of a chosen Graph Neural Network backbone (embeddings to optimize) with the

---

[*]The corresponding author.   Code: `https://github.com/allenhaozhu/COLES`.

35th Conference on Neural Information Processing Systems (NeurIPS 2021).

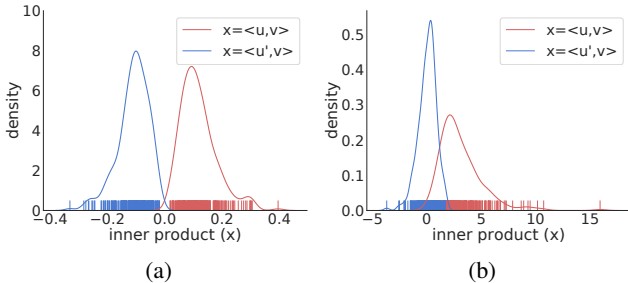

Figure 1: Densities of dot-product scores $\langle \mathbf{v}, \mathbf{u} \rangle$ and $\langle \mathbf{v}, \mathbf{u}' \rangle$ (red and blue curves) between the anchor/positive embedding and the anchor/negative embedding (GCN contrastive setting). Left/right figures use two distinct minibatches sampled on Cora. With the small overlap of distributions, many contrastive methods relying on the JS divergence may underperform (see Section 4.1 for details).

feature dimension $d'$, $\mathbf{\Theta}$ denotes network parameters, whereas $\Delta \mathbf{W} \in \mathbb{S}^n_+$ is the difference between the degree-normalized positive and negative adjacency matrices which represent the data graph and some negative graph capturing negative links for contrastive learning. Moreover, $\beta \geq 0$ controls the regularization term $\Omega(\cdot)$ whose role is to constrain the $\ell_2$ norm of network outputs or encourage the so-called incoherence [36] between column vectors. Section 3.1 presents COLES for the Linear Graph Network (LGN) family, in which we take special interest due to their simplicity and agility.

By building upon previous studies [28, 2, 48], we show that COLES can be derived by reformulating SampledNCE into Wasserstein GAN using a GAN-inspired contrastive formulation. This result has a profound impact on the performance of COLES, as the standard contrastive approaches based on SampledNCE strategy (*i.e.*, GraphSAGE [15]) turn out to utilize the Jensen-Shannon divergence, which yields $\log 2$ constant and vanishing gradients for disjoint distributions of positive and negative sampled pairs used for contrastive learning. Figure 1 shows two examples of such nearly disjoint distributions. In contrast, COLES by design avoids the sigmoid in favour of the Radial Basis Function (RBF) non-linearity. We show that such a choice coincides with a surrogate of Wasserstein distance, which is known for its robustness under poor overlap of distributions, leading to the good performance of COLES. Moreover, we also show that the loss of COLES belongs to the family of so-called block-contrastive losses, which were shown to be superior compared to pair-wise losses [3]. In summary, our contributions are threefold:

i. We derive COLES, a reformulation of the Laplacian Eigenmaps into a contrastive setting, based on the SampledNCE framework [33, 28].

ii. By using a formulation inspired by GAN, we show that COLES essentially minimizes a surrogate of Wasserstein distance, as opposed to the Jensen-Shannon (JS) divergence emerging in traditional contrastive learning. Specifically, by showing the Lipschitz continuous nature of our formulation, we prove that our formulation enjoys the Kantorovich-Rubinstein duality for the Wasserstein distance.

iii. We show COLES enjoys a block-contrastive loss known to outperform pair-wise losses [3].

**Novelty.** We propose a simple way to obtain contrastive parametric graph embeddings which works with numerous backbones. For instance, we obtain spectral graph embeddings by combining COLES with SGC [49] and S$^2$GC [61], which is solved by the SVD decomposition.

## 2 Preliminaries

**Notations.** Let $G = (V, E)$ be a simple, connected and undirected graph with $n = |V|$ nodes and $m = |E|$ edges. Let $i \in \{1, \cdots, n\}$ be the node index of $G$, and $d_j$ be the degree of node $j$ of $G$. Let $\mathbf{W}$ be the adjacency matrix, and $\mathbf{D}$ be the diagonal matrix containing degrees of nodes. Moreover, let $\mathbf{X} \in \mathbb{R}^{n \times d}$ denote the node feature matrix where each node $v$ is associated with a feature vector $\mathbf{x}_v \in \mathbb{R}^d$. Let the normalized graph Laplacian matrix be defined as $\mathbf{L} = \mathbf{I} - \mathbf{D}^{-1/2} \widehat{\mathbf{W}} \mathbf{D}^{-1/2} \in \mathbb{S}^n_+$, a symmetric positive semi-definite matrix. Finally, scalars and vectors are denoted by lowercase regular and bold fonts, respectively. Matrices are denoted by uppercase bold fonts.

## 2.1 Negative Sampling

SampledNCE [14, 33, 28], a contrastive learning framework, is used by numerous works [41, 15, 50]. Let $p_d(u|v)$ and $p_n(u'|v)$ be the so-called data and negative distributions given the so-called anchor node $v$, where $u$ and $u'$ denote the node for a positive and negative sample, respectively. Let $p_d(v)$ be the anchor distribution. Given some loss components $s_{\Theta}(v, u)$ and $\tilde{s}_{\Theta}(v, u')$ whose role is to evaluate the similarity for pairs $(v, u)$ and $(v, u')$, the contrastive loss is typically given as:

$$J(\boldsymbol{\Theta}) = \mathbb{E}_{v \sim p_d(v)} \left[ \mathbb{E}_{u \sim p_d(u|v)} s_{\Theta}(v, u) + \eta \mathbb{E}_{u' \sim p_n(u'|v)} \tilde{s}_{\Theta}(v, u') \right], \qquad (2)$$

where $\eta > 0$ controls the impact of negative sampling. Let $\mathbf{u} \in \mathbb{R}^{d'}$ be the embedding of the node $u$ obtained with an encoder $f_{\Theta}(\mathbf{x}_u)$ given parameters $\boldsymbol{\Theta}$, where $\mathbf{x}_u \in \mathbb{R}^d$ is the initial node feature vector. Let $\mathbf{u}' \in \mathbb{R}^{d'}$ and $\mathbf{v} \in \mathbb{R}^{d'}$ be embeddings of nodes $u'$ and $v$, accordingly. Let $s_{\Theta}(u, v) = \log \sigma(\mathbf{u}^{\top} \mathbf{v})$ and $\tilde{s}_{\Theta}(u', v) = \log(1 - \sigma(\mathbf{u}'^{\top} \mathbf{v}))$, where $\sigma(\cdot)$ is the sigmoid function. Subsequently, one obtains the contrastive objective (to be maximized), employed by LINE [41], REFINE [60], GraphSAGE [15] and many other methods according to Yang et al. [50]:

$$J(\boldsymbol{\Theta}) = \mathbb{E}_{v \sim p_d(v)} \left[ \mathbb{E}_{u \sim p_d(u|v)} \log \sigma(\mathbf{u}^{\top} \mathbf{v}) + \eta \mathbb{E}_{u' \sim p_n(u'|v)} \log \sigma(-\mathbf{u}'^{\top} \mathbf{v}) \right]. \qquad (3)$$

In what follows, we argue that the choice of sigmoid for $\sigma(\cdot)$ leads to negative consequences. Thus, we derive COLES under a different choice of $s_{\Theta}(v, u)$ and $\tilde{s}_{\Theta}(v, u')$.

# 3 Methodology

In what follows, we depart from the above setting of (typical) contrastive sampling, which results in a derivation of our COntrastive Laplacian EigenmapS (COLES).

## 3.1 Contrastive Laplacian Eigenmaps

Instead of log-sigmoid used in $s_{\Theta}(v, u)$ and $\tilde{s}_{\Theta}(v, u')$ of Eq. (3), let us substitute $s_{\Theta}(v, u) = \log \exp(\mathbf{u}^{\top} \mathbf{v}) = \mathbf{u}^{\top} \mathbf{v}$ and $\tilde{s}_{\Theta}(v, u') = \log \exp(-\mathbf{u}'^{\top} \mathbf{v}) = -\mathbf{u}'^{\top} \mathbf{v}$ into Eq. (2), which yields:

$$J(\boldsymbol{\Theta}) = \mathbb{E}_{v \sim p_d(v)} \left[ \mathbb{E}_{u \sim p_d(u|v)} (\mathbf{u}^{\top} \mathbf{v}) + \eta \mathbb{E}_{u' \sim p_n(u'|v)} \left( -\mathbf{u}'^{\top} \mathbf{v} \right) \right]. \qquad (4)$$

We assume that variables of the above objective (to maximize) can be constrained (*e.g.*, by the $\ell_2$ norms to prevent ill-posed solutions) and represented by degree-normalized adjacency matrices. Next, we cast Eq. (4) into the objective of COLES (refer to our Suppl. Material for derivations):

$$\begin{aligned}
\mathbf{Y}^* &= \underset{\mathbf{Y}, \text{ s.t. } \mathbf{Y}^{\top} \mathbf{Y} = \mathbf{I}}{\arg\min} \ \text{Tr}(\mathbf{Y}^{\top} \mathbf{L} \mathbf{Y}) - \frac{\eta'}{\kappa} \sum_{k=1}^{\kappa} \text{Tr}(\mathbf{Y}^{\top} \mathbf{L}_k^{(-)} \mathbf{Y}) \\
&= \underset{\mathbf{Y}, \text{ s.t. } \mathbf{Y}^{\top} \mathbf{Y} = \mathbf{I}}{\arg\max} \ \text{Tr}(\mathbf{Y}^{\top} \Delta \mathbf{W} \mathbf{Y}) \quad \text{where} \quad \Delta \mathbf{W} = \mathbf{W}^{(+)} - \frac{\eta'}{\kappa} \sum_{k=1}^{\kappa} \mathbf{W}_k^{(-)},
\end{aligned} \qquad (5)$$

and the rows of matrix $\mathbf{Y} \in \mathbb{R}^{n \times d'}$ contain the embedding vectors, $\mathbf{L}_k^{(-)}$ for $k = 1, \cdots, \kappa$ are randomly generated degree-normalized Laplacian matrices capturing the negative sampling, $\mathbf{L}_k^{(-)} = \mathbf{I} - \mathbf{W}_k^{(-)}$ and $\mathbf{L} = \mathbf{I} - \mathbf{W}^{(+)}$. The scalar $0 \leq \eta' \leq 1$ ensures that $\mathbf{L} - \frac{\eta'}{\kappa} \sum_{k=1}^{\eta'} \mathbf{L}_k^{(-)} \in \mathbb{S}_+^n$ (one could truncate the negative spectrum instead) and controls the impact of $\mathbf{W}_k^{(-)}$.

We note that COLES minimizes over the standard Laplacian Eigenmap while maximizing over the randomized Laplacian Eigenmap, which alleviates the lack of negative sampling in the original Laplacian Eigenmaps. However, unlike Laplacian Eigenmaps, we do not optimize over free variables $\mathbf{Y}$ but over the network parameters, as in Eq. (1) and (6). Clearly, if $\eta' = 0$ and $\mathbf{Y}$ are free variables, Eq. (5) reduces to standard Laplacian Eigenmaps [5]: $\mathbf{Y}^* = \arg\min_{\mathbf{Y}, \text{ s.t. } \mathbf{Y}^{\top} \mathbf{Y} = \mathbf{I}} \text{Tr}(\mathbf{Y}^{\top} \mathbf{L} \mathbf{Y})$.

> **COLES for Linear Graph Networks.** In what follows, we are especially interested in the lightweight family of LGNs such as SGC [49] and S$^2$GC [61] (APPNP [23] with the linear activation could be another choice) whose COLES-based objective can be reformulated as:
>
> $$\mathbf{P}^* = \underset{\mathbf{P}, \text{ s.t. } \mathbf{P}\mathbf{P}^{\top} = \mathbf{I}}{\arg\max} \ \text{Tr}(\mathbf{P}\mathbf{X}^{\top}\mathbf{F}^{\top} \Delta \mathbf{W}\mathbf{F}\mathbf{X}\mathbf{P}^{\top}). \qquad (6)$$

$\mathbf{F} \in \mathbb{R}^{n \times n}$ in Eq. (6) is the so-called spectral filter operating on the (degree-normalized) graph adjacency matrix, and $\mathbf{P} \in \mathbb{R}^{d' \times d}$ is a unitary projection matrix such that $0 < d' < d$. The solution to Eq. (6) can be readily obtained by solving the generalized eigenvalue problem $\mathbf{X}^\top \mathbf{F}^\top \Delta \mathbf{W} \mathbf{F} \mathbf{X} \mathbf{p} = \lambda \mathbf{p}$ (an SVD on a small $d \times d$ matrix $(\mathbf{X}^\top \mathbf{F}^\top \Delta \mathbf{W} \mathbf{F} \mathbf{X}) \in \mathbb{S}_+^d$). This step results in a matrix of embeddings $f_\mathbf{P}(\mathbf{X}) = \mathbf{F} \mathbf{X} \mathbf{P}^\top \in \mathbb{R}^{n \times d'}$ for supervised training. Based on given a degree-normalized graph adjacency matrix $\mathbf{W} \in \mathbb{R}^{n \times n}$, the spectral filters for SGC and S$^2$GC are given as $\mathbf{W}^{K'}$ and $\alpha \mathbf{I} + \frac{1-\alpha}{K'} \sum_{k=1}^{K'} \mathbf{W}^k$. Here, integer $K' \geq 1$ and scalar $\alpha \geq 0$ are the number of layers and the importance of self-loop. Note that Eq. (6) is related to Locality Preserving Projections [17] if $\eta' = 0$. Note also that enforcing the orthogonality constraints in Eq. (6) coincides with the SVD-based solution described above. In contrast, the more general form of COLES in Eq. (1) requires the regularization or constraints (depending on the backbone) imposed on minibatches $i \in \mathcal{B}$ e.g., we used the soft penalty $\Omega(f_\Theta(\mathbf{X}_i)) = \|f_\Theta^\top(\mathbf{X}_i) f_\Theta(\mathbf{X}_i) - \mathbf{I}\|_F^2$.

> **COLES (Stiefel).** Inspired by the Locality Preserving Projections [17] and Eq. (6), we also investigate:
> $$(\mathbf{P}^*, \Theta^*) = \underset{\mathbf{P}, \Theta, \text{ s.t. } \mathbf{P}\mathbf{P}^\top = \mathbf{I}}{\arg\max} \operatorname{Tr}(\mathbf{P} f_\Theta^\top(\mathbf{X}) \Delta \mathbf{W} f_\Theta(\mathbf{X}) \mathbf{P}^\top), \tag{7}$$
> solved on the Stiefel manifold by GeoTorch [29]. The embed. is: $f_\mathbf{P}(\mathbf{X}) = f_\Theta(\mathbf{X}) \mathbf{P}^\top \in \mathbb{R}^{n \times d'}$.

## 4 Theoretical Analysis

### 4.1 COLES is Wasserstein-based Contrastive Learning

By casting the positive and negative distributions of SampledNCE as the real and generated data distributions of GAN, the key idea of this analysis is to (i) cast the traditional contrastive loss in Eq. (3) (used by LINE [41], GraphSAGE [15] and other methods [50]) as a GAN framework, and show this corresponds to the use of JS divergence and (ii) cast the objective of COLES in Eq. (4) as a GAN framework, and show it corresponds to the use of a surrogate of Wasserstein distance. The latter outcome is preferable under the vanishing overlap of two distributions, as the JS divergence yields $\log(2)$ constant and vanishing gradients. The Wasserstein distance suffer less from this issue.

For simplicity, consider the embedding $\mathbf{v}$ of the anchor node is given. An embedding vector $\mathbf{u}$ is sampled from the 'real' distribution $p_r(\mathbf{u}) = p_d(u \mid v)$, and $\mathbf{u}'$ is sampled from the 'generator' distribution $p_g(\mathbf{u}) = p_n(u' \mid v)$. Following Arjovsky et al. [2] and Weng [48], one arrives at a GAN-inspired formulation which depends on the choice of 'discriminator' $D(\mathbf{u})$:

$$\max_\Theta \int_\mathbf{u} \left( p_r(\mathbf{u}) \log(D(\mathbf{u})) + p_g(\mathbf{u}) \log(1 - D(\mathbf{u})) \right) d\mathbf{u} \leq 2\operatorname{JS}(p_r \| p_g) - 2\log 2, \tag{8}$$

where $\operatorname{JS}(p_r \| p_g)$ denotes the Jensen-Shannon (JS) divergence. If $D(\mathbf{u})$ is completely free, then the optimal $D^*(\mathbf{u})$ which maximizes the left-hand-side (LHS) of Eq. (8) is $D(\mathbf{u}) = p_r(\mathbf{u})/(p_r(\mathbf{u}) + p_g(\mathbf{u}))$. Plugging $D^*$ back into the LHS, we get the right-hand-side (RHS) of the inequality. In our setting, the case $p_g \sim p_r$ means that negative sampling yields hard negatives, that is, negative and positive samples are very similar. Hence, this family of embedding techniques try to optimally discriminate $p_r$ and $p_g$ in the embedding space.

The above analysis shows that traditional contrastive losses are bounded by the JS divergence. Regardless of the choice of $D(\mathbf{u})$, if the support of the density $p_r$ and the support of $p_g$ are disjoint (e.g., positive and negative samples in the minibatch of the SGD optimization), the JS divergence yields zero and vanishing gradients. If the 'discriminator' is set to $D(\mathbf{u}) = \sigma(\mathbf{u}^\top \mathbf{v})$, the objective in Eq. (8) becomes exactly Eq. (3). By noting $\partial \log \sigma(\mathbf{u}^\top \mathbf{v})/\partial \mathbf{u} = \sigma(\mathbf{u}^\top \mathbf{v})\mathbf{v}$, the gradient is likely to vanish due to the scalar $\sigma(\mathbf{u}^\top \mathbf{v})$ and does not contribute to learning of network parameters. Figure 1 shows densities of $x = \mathbf{u}^\top \mathbf{v}$ and $x = \mathbf{u}'^\top \mathbf{v}$ for $p_r$ and $p_d$ estimated by the Parzen window on two sampled minibatches of contrastive GCN. Clearly, these distributions are approximately disjoint.

Compared with the JS divergence, the Wasserstein distance considers the metric structure of the embedding space:
$$\inf_{\gamma \sim \Pi(p_r, p_g)} \mathbb{E}_{(\mathbf{u}, \mathbf{u}') \sim \gamma} \|\mathbf{u} - \mathbf{u}'\|_1, \tag{9}$$

where $\Pi(p_r, p_g)$ is the set of joint distributions with marginals $p_r(\mathbf{u})$ and $p_g(\mathbf{u}')$.

By the Kantorovich-Rubinstein duality [45], **the optimal transport problem for COLES** can be equivalently expressed as:

$$
\sup_{g:\,K(g)\leq 1} \left( \mathbb{E}_{\mathbf{u}\sim p_r}[g(\mathbf{u})] - \mathbb{E}_{\mathbf{u}'\sim p_g}[g(\mathbf{u}')] \right)
$$
$$
\geq \max_{\boldsymbol{\Theta}} \left[ \mathbb{E}_{u\sim p_d(u|v)}(\mathbf{u}^\top \mathbf{v}) + \mathbb{E}_{u'\sim p_n(u'|v)}(-\mathbf{u}'^\top \mathbf{v}) \right], \tag{10}
$$

under a drawn anchor $v \sim p_d(v)$, where $K(g)$ means the Lipschitz constant, and supreme is taken over all 1-Lipschitz functions (or equivalently, all $K$-Lipschitz functions.)

The "$\geq$" is because $g(\mathbf{u})$ is chosen to the specific form $g_v(\mathbf{u}) = \mathbf{u}^\top \mathbf{v}$, where $\mathbf{v}$ is parameterized by a graph neural network with parameters $\boldsymbol{\Theta}$. Optimizing over the neural network parameters $\boldsymbol{\Theta}$ can enumerate a subset of functions which satisfies the Lipschitz constant $K$.

**Lipschitz continuity of COLES.** In order to assure the Lipschitz continuity of COLES, let individual embeddings be stacked row-wise into a matrix and $\ell_2$-norm normalized along rows, or along columns. Given $v$ (the reference node), the following holds:

$$
|\mathbf{u}^\top \mathbf{v} - \mathbf{u}'^\top \mathbf{v}| \leq \|\mathbf{v}\|_{\max}\|\mathbf{u} - \mathbf{u}'\|_1,
$$

where $K = \max_v \|\mathbf{v}\|_{\max}$ ($\leq 1$ in the case of either sphere embedding or the constraint $\mathbf{Y}^\top \mathbf{Y} = \mathbf{I}$ of the COLES formula in Eq. (5)). Thus, the function $g(\mathbf{u}) = \mathbf{u}^\top \mathbf{v}$ is Lipschitz with constant $K$.

## 4.2 COLES enjoys the Block-contrastive Loss

We notice that COLES leverages an access to blocks of similar data, rather than just individual pairs in the loss function. To this end, we resort to the Prop. 6.2 of Arora et al. [3], which shows that for family of functions $\mathcal{F}$ whose $\|f(\cdot)\| \leq R$ for some $R > 0$, a block-contrastive loss $L_{un}^{\text{block}}$ is always bounded by a pairwise-contrastive loss $L_{\text{un}}$, that is, $L_{un}^{\text{block}}(f) \leq L_{\text{un}}(f)$. To that end, Arora et al. [3] also show that as block-contrastive losses achieve lower minima than their pairwise-contrastive counterparts, they also enjoy better generalization.

We show that COLES is a block-contrastive loss, which explains its good performance. Following Eq. (4), for a given embedding $\mathbf{v} = f_{\boldsymbol{\Theta}}(\mathbf{x}_v)$, and $b$ embeddings $\mathbf{u}_i = f_{\boldsymbol{\Theta}}(\mathbf{x}_{u_i})$ and $\mathbf{u}'_i = f_{\boldsymbol{\Theta}}(\mathbf{x}_{u'_i})$ drawn according to $p_d(u \mid v)$ and $p_n(u' \mid v)$, we have (note minus preceding eq. as here we minimize):

$$
-\mathbb{E}_{u\sim p_d(u|v)}(\mathbf{u}^\top \mathbf{v}) + \mathbb{E}_{u'\sim p_n(u'|v)}\left(-\mathbf{u}'^\top \mathbf{v}\right) = -\mathbf{v}^\top \left( \frac{\sum_i \mathbf{u}_i}{b} - \frac{\sum_i \mathbf{u}'_i}{b'} \right) = -\mathbf{v}^\top(\boldsymbol{\mu}^+ - \boldsymbol{\mu}^-), \tag{11}
$$

where $\boldsymbol{\mu}^+$ and $\boldsymbol{\mu}^-$ are positive and negative block summaries of sampled nodes. Looking at Eq. (5), it is straightforward to simply expand $\sum_{(i,j)\in E} \|\mathbf{y}_i - \mathbf{y}_j\|_2^2 \Delta W_{ij}$ to see that each index $i$ will act as a selector of anchors, whereas index $j$ will loop over positive and negative samples taking into account their connectivity to $i$ captured by $\Delta W_{ij}$. We provide this expansion in the Suppl. Material.

## 4.3 Geometric Interpretation.

Below, we analyze COLES through the lens of *Alignment and Uniformity on the Hypersphere* of Wang and Isola [47]. To this end, we decompose our objective into the so-called alignment and uniformity losses. Firstly, Mikolov et al. [33] have shown that SampledNCE with the sigmoid non-linearity is a practical approximation of SoftMax contrastive loss, the latter suffering poor scalability w.r.t. the count of negative samples. For this reason, many contrastive approaches (DeepWalk, GraphSAGE, DGI, Graph2Gauss, *etc*.) adopt SampledNCE rather than SoftMax (GRACE) framework.

Wang and Isola [47] have decomposed the SoftMax contrastive loss into $\mathcal{L}_{align}$ and $\mathcal{L}_{umiform}$ [47]:

$$
\mathcal{L}(u, v, \mathcal{N}) = \mathcal{L}_{\text{align}}(u, v) + \mathcal{L}_{\text{uniform}}(u', v, \mathcal{N}) = -\log \frac{e^{\mathbf{u}^\top \mathbf{v}}}{e^{\mathbf{u}^\top \mathbf{v}} + \sum_{u'\in\mathcal{N}} e^{\mathbf{u}'^\top \mathbf{v}}} \tag{12}
$$

where $\mathcal{N}$ is a sampled subset of negative samples, $u$ and $v$ are node indexes of so-called positive sample and anchor embeddings, $\mathbf{u}$ and $\mathbf{v}$. Let $\langle \mathbf{u}, \mathbf{u} \rangle = \langle \mathbf{u}', \mathbf{u}' \rangle = \langle \mathbf{v}, \mathbf{v} \rangle = \tau^2$ ($\tau$ acts as the so-called temperature). Moreover, $\mathcal{L}_{\text{align}} = -\langle \mathbf{u}, \mathbf{v} \rangle$ and $\mathcal{L}_{\text{uniform}} = \log \sum_{u^\ddagger \in \mathcal{N}\cup\{u\}} e^{\mathbf{u}^{\ddagger\top}\mathbf{v}}$, that is, $\mathcal{L}_{\text{uniform}}$ is

a logarithm of an arithmetic mean of RBF responses over the subset $\mathcal{N} \cup \{u\}$. Of course, computing the total loss $\mathcal{L}$ requires drawing $u$ and $v$ from the graph and summing over multiple $\mathcal{L}_{align}(u, v)$ and $\mathcal{L}'_{uniform}(u, v, \mathcal{N})$ but we skip this step and the argument variables of loss functions for brevity.

COLES can be decomposed into $\mathcal{L}_{align}$ and $\mathcal{L}_{umiform}$ [47] as follows:

$$\mathcal{L}_{\text{align}} + \mathcal{L}'_{\text{uniform}} = -\log e^{\mathbf{u}^\top \mathbf{v}} - \frac{1}{|\mathcal{N}|} \sum_{u' \in \mathcal{N}} \log e^{-\mathbf{u}'^\top \mathbf{v}} = -\log \frac{e^{\mathbf{u}^\top \mathbf{v}}}{\left(\Pi_{u' \in \mathcal{N}} e^{\mathbf{u}'^\top \mathbf{v}}\right)^{\frac{1}{|\mathcal{N}|}}}, \qquad (13)$$

where $\mathcal{L}_{\text{align}}$ remains the same with SoftMax but $\mathcal{L}'_{\text{uniform}} = \log \left(\Pi_{u' \in \mathcal{N}} e^{\mathbf{u}'^\top \mathbf{v}}\right)^{\frac{1}{|\mathcal{N}|}}$ is in fact a logarithm of the geometric mean of RBF responses over the subset $\mathcal{N}$. Thus, our loss can be seen as the ratio of geometric means over RBF functions. Several authors (*e.g.*, Gonzalez [12]) noted that the geometric mean helps smooth out the Gaussian noise under the i.i.d. uniform sampling while loosing less information than the arithmetic mean. The geometric mean enjoys better confidence intervals the arithmetic mean given a small number of samples. As we sample few negative nodes for efficacy, we expect the geometric mean is more reliable. Eq. (12) and (13) are just two specific cases of a generalized loss:

$$\mathcal{L}_{\text{align}} + \mathcal{L}''_{\text{uniform}} = -\log \frac{e^{\mathbf{u}^\top \mathbf{v}}}{M_p \left(e^{\mathbf{u}'_1{}^\top \mathbf{v}}, \cdots, e^{\mathbf{u}'_{|\mathcal{N}|}{}^\top \mathbf{v}}\right)}, \qquad (14)$$

where $M_p(\cdot)$ in $\mathcal{L}''_{\text{uniform}} = \log M_p \left(e^{\mathbf{u}'_1{}^\top \mathbf{v}}, \cdots, e^{\mathbf{u}'_{|\mathcal{N}|}{}^\top \mathbf{v}}\right)$ is the so-called generalized mean. We introduce $M_p(\cdot)$ into the denominator of Eq. (14) but it can be also introduced in the numerator. We investigate the geometric ($p{=}0$), arithmetic ($p{=}1$), harmonic ($p{=}-1$) and quadratic ($p{=}2$) means.

## 5 Related Works

**Graph Embeddings.** Graph embedding methods such as Laplacian Eigenmaps [5] and IsoMap [42] reduce the dimensionality of representations by assuming the data lies on a low-dimensional manifold. With these methods, for a set of high-dimensional data features, a similarity graph is built based on the pairwise feature similarity, and each node embedded into a low-dimensional space. The graph is constructed from non-relational high dimensional data features, and Laplacian Eigenmaps ignore relations between dissimilar node pairs, that is, embeddings of dissimilar nodes are not penalized.

To alleviate the above shortcomings, DeepWalk [35] uses truncated random walks to explore the network structure and utilizes the skip-gram model [32] for word embedding to derive the embedding vectors of nodes. LINE [41] explores a similar idea with an explicit objective function by setting the walk length as one and applying negative sampling [32]. Node2Vec [13] interpolates between breadth- and depth-first sampling strategies to aggregate different types of neighborhoods.

**Representation Learning for Graph Neural Networks.** Supervised and (semi-)supervised GNNs [22] require labeled datasets that may not be readily available. Yet, unsupervised GNNs have received little attention. GCN [22] employs the minimization of reconstruction error as the objective function to train the encoder. GraphSAGE [15] incorporates objectives inspired by Deep-Walk *e.g.*, contrastive loss encouraging nearby nodes to have similar representations while preserving dissimilarity between representations of disparate nodes. DGI [44], inspired by Deep InfoMax (DIM) [18], proposes an objective with global-local sampling strategy, which maximizes the Mutual Information (MI) between global and local graph embeddings. In contrast, Augmented Multiscale Deep InfoMax (AMDIM) [4] maximizes MI between multiple views of data. MVRLG [16] contrasts encodings from first-order neighbors and a graph diffusion. MVRLG uses GCNs to learn node embeddings for different views. Fisher-Bures Adversary GCN [40] assumes that the graph is generated w.r.t. some observation noise. Graph-adaptive ReLU [58] uses an adaptive non-linearity in GCN. Multi-view augmentation-based methods, not studied by us, are complementary to COLES. Moreover, linear networks *e.g.*, SGC [49] and $\text{S}^2\text{GC}$ [61] capture the neighborhood and increasingly larger neighborhoods of each node, respectively. SGC and $\text{S}^2\text{GC}$ have no projection layer, which results in embeddings of size equal to the input dimension. DGI [44] uses the block-contrastive strategy [3] by treating negative samples as a difference of instances and a summary of node embeddings for positive samples. Finally, COLES can be extended to other domains/problems *e.g.*, time series/change point detection [9] or few-shot learning [39, 54, 55].

**(Negative) Sampling.** Sampling node pairs relies on random walks [35] or second-order proximity [41], *etc*. In contrast, COLES samples an undirected graph based on the random graph sampling theory [11], where each edge is independently chosen with a prescribed probability $p' > 0$.

# 6 Experiments

We evaluate COLES on transductive and inductive node classification tasks. Node clustering is also evaluated. COLES is compared to state-of-the-art unsupervised, contrastive and (semi-)supervised methods. Unsupervised methods do not use label information except for the classifier. Contrastive methods use the contrastive setting to learn similarity/dissimilarity. (Semi-)supervised methods use labels to train their projection layer and classifier. By *semi-supervised*, we mean that only a few of nodes used for training are labeled. (Semi-)supervised models use a SoftMax classifier, whereas unsupervised and contrastive methods use a logistic regression classifier.

**Datasets.** COLES is evaluated on four citation networks: Cora, Citeseer, Pubmed, Cora Full [22, 6] for transductive setting. We also employ the large scale Ogbn-arxiv from OGB [19]. Finally, the Reddit [53] dataset is used in inductive setting. Table 1 provides details of all datasets.

**Metrics.** Fixed data splits [51] for transductive tasks are often used in evaluations between different models. However, such an experimental setup may benefit easily overfitting models [38]. Thus, instead of fixed data splits, results are averaged over 50 random splits for each dataset and standard deviations are reported for empirical evaluation on transductive tasks. Moreover, we also test the performance under a different number of samples per class *i.e.*, 5 and 20 samples per class. Typically, the performance for the inductive task is tested on relatively larger graphs. Thus, we choose fixed data splits as in previous papers [15, 53], and we report the Micro-F1 scores averaged on 10 runs.

**Baseline models.** We group baseline models into unsupervised, contrastive and (semi-)supervised methods, and implement them in the same framework/testbed. Contrastive methods include Deep-Walk [35], GCN+SampledNCE developed as an alternative to GraphSAGE+SampledNCE [15], Graph2Gauss [6], SCE [56], DGI [44], GRACE [62], GCA [63] and GraphCL [52], which are our main competitors. Note that GRACE, GCA and GraphCL are based on multi-view and data augmentation, and GraphCL is mainly intended for graph classification. We do not study graph classification as it requires advanced node pooling [24] with mixed- or high-order statistics [26, 25, 27]. We compare results with representative (semi-)supervised GCN [22], GAT [44] and MixHop [1] models. SGC and $S^2GC$ are unsupervised spectral filter networks. They do not have any learnable parameters that depend on labels, with exception of a classifier. To reduce the resulting dimensionality, we also add PCA-$S^2GC$ and RP-$S^2GC$, which use PCA and random projections to obtain the projection layer on these methods. We extend our COLES framework with different GNNs: GCN, SGC and/or $S^2GC$, and we name them COLES-GCN, COLES-SGC and COLES-$S^2GC$. As COLES-GCN is a multi-layer non-linear encoder, the optimization of COLES-GCN is non-convex. The optimization of COLES-SGC and COLES-$S^2GC$ is convex if $\mathbf{L} - \frac{\eta'}{\kappa} \sum_{k=1}^{\eta'} \mathbf{L}_k^{(-)} \in \mathbb{S}_+^n$, and COLES-GCN (Stiefel) is convex w.r.t. $\mathbf{P}$. We set hyperparameters based on the settings described in their papers.

**General model setup.** For all (semi-)supervised models, we use early stopping on each random split and we capture the corresponding classification result. For all unsupervised models, we choose the embedding dimension to be 512 on Cora, Citeseer and Cora Full, and 256 on Pubmed. After the embeddings of nodes are learnt, a classifier is trained by applying the logistic regression in the embedding space. For inductive learning, methods based on COLES use 512-dimensional embeddings. Other hyperparameters for the baseline models are the same as in original papers.

**Hyperparameter of our models.** In the transductive experiments, the detailed hyperparameter settings for Cora, Citeseer, Pubmed, and Cora Full are listed below. For COLES, we use the Adam optimizer with learning rates of $[0.001, 0.0001, 0.02, 0.02]$ and the decay of $[5e–4, 1e–3, 5e–4, 2e–4]$. The number of training epochs are $[20, 20, 100, 30]$, respectively. We sample 10 randomized adjacent matrices, and 5 negative samples for each node in each matrix on each dataset before training. For the $S^2GC$ and COLES-$S^2GC$, the number of propagation steps (layers) are 8 for all datasets except Cora Full (2 steps). For SGC and COLES-SGC, we use 2 steps for all datasets.

## 6.1 Transductive Learning

In this section, we consider transductive learning where all nodes are available in the training process.

**Contrastive Embedding Baselines *vs*. COLES.** Table 2 shows that the performance of COLES-GCN and the linear variant, COLES-$S^2GC$, are better than other unsupervised models. In particular, COLES-GCN outperforms GCN+SampledNCE on all four datasets, which shows that COLES has an advantage over the SampledNCE framework. In addition, COLES-$S^2GC$ typically outperforms the

Table 1: The statistics of datasets.

| Dataset | Task | Nodes | Edges | Features | Classes |
|---------|------|-------|-------|----------|---------|
| Cora | Transductive | 2,708 | 5,429 | 1,433 | 7 |
| Citeseer | Transductive | 3,327 | 4,732 | 3,703 | 6 |
| Pubmed | Transductive | 19,717 | 44,338 | 500 | 3 |
| Cora Full | Transductive | 19,793 | 65,311 | 8,710 | 70 |
| Ogbn-arxiv | Transductive | 169,343 | 1,166,243 | 128 | 40 |
| Reddit | Inductive | 232,965 | 11,606,919 | 602 | 41 |

best contrastive baseline DGI by up to 3.4%. In Cora Full, we notice that $S^2GC$ underperforms when training with 5 samples. However, COLES-$S^2GC$ is able to significantly boost its performance by 9%. On Citeseer with 5 training samples, COLES-$S^2GC$ outperforms $S^2GC$ by 6.8%. We also note that COLES-GCN (Stiefel) outperforms COLES-GCN (based on the soft-orthogonality constraint) by up to 2.7% but its performance below the performance of COLES-$S^2GC$.

Noteworthy is that for augmentation-based methods, COLES-GCN with augmentations denoted as COLES-GCN (+Aug) outperforms COLES-GCN without augmentations. COLES-GCN (+Aug) also outperforms GRACE and GCA, and GraphCL in most experiments. Nonetheless, COLES-$S^2GC$ without any augmentations outperformed all augmentation-based methods.

Finally, Table 7 shows that COLES-$S^2GC$ outperforms all other methods on the challenging Ogbn-arxiv, while using a very small number of trainable parameters.

**Semi-supervised GNNs *vs*. COLES.** Table 2 shows that the contrastive GCN baselines perform worse than semi-supervised variants, especially when 20 labeled samples per class are available. In contrast, COLES-GCN outperformed the semi-supervised GCN on Cora by 6.3% and 1.4% given 5 and 20 labeled samples per class. COLES-GCN also outperforms GCN on Citeseer and Cora Full by 8.3% and 6.3% given 5 labeled samples per class. When the number of labels per class is 5, COLES-$S^2GC$ outperforms GCN by a margin of 8.1% on Cora and 9.4% on Citeseer. These results show the superiority of COLES on four datasets when the number of samples per class is 5. Even for 20 labeled samples per class, COLES-$S^2GC$ outperforms the best semi-supervised baselines on all four datasets *e.g.*, by 1.7% on Citeseer. Semi-supervised models are affected by the low number of labeled samples, which is consistent with [31], *e.g.*, for GAT and MixHop. The accuracy of COLES-GCN and COLES-$S^2GC$ is not affected as significantly due to the contrastive setting.

**Unsupervised GNNs *vs*. COLES.** SGC and $S^2GC$ are unsupervised LGNs as they are spectral filters which do not use labels (except for the classifier). Table 2 shows that COLES-$S^2GC$ outperforms RP-$S^2GC$ and PCA-$S^2GC$ under the same size of projections. In most cases, COLES-$S^2GC$ also outperforms the unsupervised $S^2GC$ baseline (high-dimensional representation).

Table 2: Mean classification accuracy (%) and the standard dev. over 50 random splits. Numbers of labeled samples per class are in parentheses. The best accuracy per column is in bold. Models are organized into semi-supervised, contrastive and unsupervised groups. OOM means out of memory.

| | Method | Cora (5) | Cora (20) | Citeseer (5) | Citeseer (20) | Pubmed (5) | Pubmed (20) | Cora Full (5) | Cora Full (20) |
|---|--------|----------|-----------|--------------|---------------|------------|-------------|---------------|----------------|
| Semi-supervised | GCN | 67.5±4.8 | 79.4±1.6 | 57.7±4.7 | 69.4±1.4 | 65.4±5.2 | 77.2±2.1 | 49.3±1.8 | 61.5±0.5 |
| | GAT | 71.2±3.5 | 79.6±1.5 | 54.9±5.0 | 69.1±1.5 | 65.5±4.6 | 75.4±2.3 | 43.9±1.5 | 56.9±0.6 |
| | MixHop | 67.9±5.7 | 80.0±1.4 | 54.5±4.3 | 67.1±2.0 | 64.4±5.6 | 75.7±2.7 | 47.5±1.5 | 61.0±0.7 |
| Contrastive | DeepWalk | 60.3±4.0 | 70.5±1.9 | 38.3±2.9 | 45.6±2.0 | 60.3±5.6 | 70.8±2.6 | 38.9±1.4 | 51.1±0.7 |
| | GCN+SampledNCE | 61.3±4.3 | 74.3±1.6 | 42.3±3.4 | 56.8±1.9 | 60.9±5.7 | 70.3±2.5 | 32.7±1.9 | 45.2±0.9 |
| | SAGE+SampledNCE | 65.0±3.5 | 73.8±1.5 | 48.0±3.5 | 56.5±1.6 | 64.1±6.1 | 74.6±1.9 | 35.0±1.4 | 43.6±0.6 |
| | Graph2Gauss | 72.7±2.0 | 76.2±1.1 | 60.7±3.5 | 65.7±1.5 | 67.6±3.9 | 74.1±2.1 | 38.9±1.3 | 49.3±0.5 |
| | SCE | 74.3±2.7 | 80.2±1.1 | 65.4±2.9 | 70.7±1.2 | 65.7±6.0 | 75.8±2.2 | 50.7±1.5 | 60.6±0.6 |
| | DGI | 72.9±4.0 | 78.1±1.8 | 65.7±3.6 | 71.1±1.1 | 65.3±5.7 | 73.9±2.3 | 50.5±1.4 | 58.4±0.6 |
| | COLES-GCN | 73.8±3.4 | 80.8±1.3 | 66.0±2.6 | 69.0±1.3 | 62.7±4.6 | 72.7±2.1 | 47.3±1.5 | 58.9±0.5 |
| | COLES-GCN (Stiefel) | 75.0±3.4 | 81.0±1.3 | **67.9±2.3** | **71.7±0.9** | 62.6±5.0 | 73.2±2.6 | 47.6±1.2 | 59.2±0.5 |
| | COLES-$S^2GC$ | **76.5±2.6** | **81.5±1.2** | 67.5±2.2 | 71.3±1.0 | 66.0±5.2 | **77.4±1.9** | **50.8±1.4** | **61.8±0.5** |
| Contrastive + Augmentation | GraphCL | 72.6±4.2 | 78.3±1.7 | 65.6±3.0 | 71.1±0.8 | OOM | OOM | OOM | OOM |
| | GRACE | 64.9±4.2 | 73.9±1.6 | 61.8±3.9 | 68.4±1.6 | OOM | OOM | OOM | OOM |
| | GCA | 61.5±4.9 | 75.8±1.9 | 43.2±3.6 | 55.7±1.9 | OOM | OOM | OOM | OOM |
| | COLES-GCN (+Aug) | 75.3±3.3 | 81.0±1.3 | 66.7±2.3 | 69.8±1.3 | 63.9±5.0 | 73.4±2.5 | 48.0±1.2 | 59.4±0.5 |
| Unsupervised | SGC | 63.9±5.4 | 78.3±1.9 | 59.5±3.4 | 69.8±1.4 | 65.8±4.4 | 76.3±2.3 | 46.0±2.2 | 57.7±1.2 |
| | $S^2GC$ | 71.4±4.4 | 81.3±1.2 | 60.3±4.0 | 69.5±1.2 | **67.6±4.2** | 73.3±2.0 | 41.8±1.7 | 60.0±0.5 |
| | PCA-$S^2GC$ | 72.1±3.8 | 81.2±1.3 | 61.0±3.5 | 68.8±1.3 | 67.5±4.3 | 73.2±2.0 | 42.3±1.7 | 59.3±0.6 |
| | RP-$S^2GC$ | 65.9±4.6 | 78.1±1.2 | 51.4±3.2 | 61.7±1.6 | 66.1±5.0 | 72.5±1.9 | 31.5±1.4 | 48.7±0.6 |

Table 3: The influence of the number ($\kappa$) of negative Laplacian graphs on COLES-S$^2$GC. Parentheses show the no. of labeled samples p/class.

| | | $\kappa \to 0$ | 1 | 5 | 10 |
|---|---|---|---|---|---|
| Cora | (20) | 79.88 | 81.43 | 81.18 | 81.17 |
| Cora | (5) | 70.12 | 76.24 | 75.89 | 75.79 |
| Citeseer | (20) | 69.42 | 70.71 | 70.61 | 70.61 |
| Citeseer | (5) | 58.17 | 67.03 | 66.96 | 67.04 |

Table 4: The influence of the number ($\kappa$) of negative Laplacian graphs on COLES-GCN. Parentheses show the no. of labeled samples p/class.

| | | $\kappa \to 0$ | 1 | 5 | 10 |
|---|---|---|---|---|---|
| Cora | (20) | 75.70 | 80.90 | 80.87 | 80.90 |
| Cora | (5) | 60.97 | 74.14 | 74.11 | 74.07 |
| Citeseer | (20) | 60.61 | 69.04 | 69.21 | 69.08 |
| Citeseer | (5) | 45.31 | 65.85 | 66.08 | 66.01 |

Table 5: The performance given various choices of the generalized mean $M_p$ for the uniformity loss.

| Method | Cora | | Citeseer | | Pubmed | | Cora Full | |
|---|---|---|---|---|---|---|---|---|
| | (5) | (20) | (5) | (20) | (5) | (20) | (5) | (20) |
| Geometric ($M_0$) (COLES-S$^2$GC) | 76.5±2.6 | 81.5±1.2 | 67.5±2.2 | 71.3±1.0 | 66.0±5.2 | 77.4±1.9 | 50.8±1.4 | 61.8±0.5 |
| Arithmetic ($M_1$) (SoftMax-Contr.) | 71.8±3.0 | 77.6±1.3 | 63.2±3.1 | 69.3±0.8 | 65.9±4.3 | 77.1±1.5 | 49.2±1.4 | 60.6±0.6 |
| Harmonic ($M_{-1}$) | 75.2±3.5 | 80.7±1.2 | 64.7±2.4 | 70.9±0.9 | 65.9±5.5 | 73.9±2.4 | 48.0±1.6 | 59.7±1.6 |
| Quadratic ($M_2$) | 72.3±2.5 | 77.2±1.3 | 65.4±2.2 | 70.7.±0.8 | 65.6±4.5 | 77.3±1.5 | 49.2±1.5 | 60.6±1.6 |

**Negative Laplacian Eigenmaps.** Below, we analyze how $\kappa$ in Eq. (5) influences the performance. We set $\kappa \in \{0, 1, 5, 10\}$ on COLES-S$^2$GC and COLES-GCN given Cora and Citeseer with 5 and 20 labeled samples per class. The case of $\kappa = 0$ means no negative Laplacian Eigenmaps are used, thus the solution simplifies to regular Laplacian Eigenmaps parametrized by GCN embeddings. Table 3 shows that without the negative Laplacian Eigenmaps, the performance of COLES-S$^2$GC drops significantly *i.e.*, between 6% and 9% for 5 labeled samples per class. That means the negative Laplacian Eigenmaps play important role which highlights the benefits of COLES. Although negative Laplacian Eigenmaps improve results, using $\kappa > 1$ negative matrices improves the performance only marginally. Table 4 shows that COLES-GCN relies on negative Laplacian Eigenmaps. Without negative Laplacian Eigenmaps, the performance of COLES-GCN drops by 20% on Citeseer with 5 samples per class. Even when 20 samples per class are used, if $\kappa = 0$, the performance of COLES-GCN drops by 8.4%.

## 6.2 Uniformity Loss as the Generalized Mean ($M_p$).

Following the analysis presented in Section 4.3, Table 5 demonstrates the impact of the choices of the uniformity loss on the performance of COLES. To this end, we select the geometric ($M_0$), arithmetic ($M_1$), harmonic ($M_{-1}$) and quadratic($M_2$) means as examples of $M_p$ realizing the uniformity loss. On all the investigated datasets, the geometric mean outperforms other variants.

## 6.3 Inductive Learning

In inductive learning, models have no access to the test set, thus they need to generalize well to unseen samples. Table 6 shows that COLES enjoys a significant performance gain (1% - 5% in Micro-F1 scores), performing close to supervised methods with a low memory footprint. In contrast, DGI on Reddit triggers out-of-memory errors on Nvidia GTX 1080 GPU (94.0 Micro-F1 is taken from [44]).

Table 6: Test Micro F1 Score (%) averaged over 10 runs on Reddit. Results of other models are from original papers.

| Setting | Model | Test F1 |
|---|---|---|
| Supervised | SAGE-mean | 95.0 |
| | SAGE-LSTM | **95.4** |
| | SAGE-GCN | 93.0 |
| Contrastive | SAGE-mean | 89.7 |
| | SAGE-LSTM | 90.7 |
| | SAGE-GCN | 90.8 |
| | FastGCN | 93.7 |
| | DGI | 94.0 |
| | COLES-GCN | 94.0 |
| | COLES-SGC | 94.8 |
| | COLES-S$^2$GC | **95.4** |

Table 7: Mean classification accuracy (%) and the standard dev. over 10 runs on Ogbn-arxiv. Results of other models are from original papers.

| Method | | Test Acc. | #Params |
|---|---|---|---|
| MLP | | 55.50±0.23 | 110,120 |
| Node2Vec | [13] | 70.07±0.13 | 21,818,792 |
| GraphZoom | [10] | 71.18±0.18 | 8,963,624 |
| C&S | [20] | 71.26±0.01 | 5,160 |
| SAGE-mean | [15] | 71.49±0.27 | 218,664 |
| GCN | [22] | 71.74±0.29 | 142,888 |
| DeeperGCN | [30] | 71.92±0.17 | 491,176 |
| SIGN | [37] | 71.95±0.11 | 3,566,128 |
| FrameLet | [59] | 71.97±0.12 | 1,633,183 |
| S$^2$GC | [61] | 72.01±0.25 | 110,120 |
| COLES-S$^2$GC | | **72.48±0.25** | 110,120 |

Table 8: The clustering performance on Cora, Citeseer and Pubmed.

| Method | Input | Cora | | | Citeseer | | | Pubmed | | |
|--------|-------|------|------|------|----------|------|------|--------|------|------|
| | | Acc% | NMI% | F1% | Acc% | NMI% | F1% | Acc% | NMI% | F1% |
| k-means | Feature | 34.65 | 16.73 | 25.42 | 38.49 | 17.02 | 30.47 | 57.32 | 29.12 | 57.35 |
| Spectral-f | Feature | 36.26 | 15.09 | 25.64 | 46.23 | 21.19 | 33.70 | 59.91 | 32.55 | 58.61 |
| Spectral-g | Graph | 34.19 | 19.49 | 30.17 | 25.91 | 11.84 | 29.48 | 39.74 | 3.46 | 51.97 |
| DeepWalk | Graph | 46.74 | 31.75 | 38.06 | 36.15 | 9.66 | 26.70 | 61.86 | 16.71 | 47.06 |
| GAE | Both | 53.25 | 40.69 | 41.97 | 41.26 | 18.34 | 29.13 | 64.08 | 22.97 | 49.26 |
| VGAE | Both | 55.95 | 38.45 | 41.50 | 44.38 | 22.71 | 31.88 | 65.48 | 25.09 | 50.95 |
| ARGE | Both | 64.00 | 44.90 | 61.90 | 57.30 | 35.00 | 54.60 | 59.12 | 23.17 | 58.41 |
| ARVGE | Both | 62.66 | 45.28 | 62.15 | 54.40 | 26.10 | 52.90 | 58.22 | 20.62 | 23.04 |
| GCN | Both | 59.05 | 43.06 | 59.38 | 45.97 | 20.08 | 45.57 | 61.88 | 25.48 | 60.70 |
| SGC | Both | 62.87 | 50.05 | 58.60 | 52.77 | 32.90 | 63.90 | 69.09 | 31.64 | 68.45 |
| S$^2$GC | Both | 68.96 | 54.22 | **65.43** | 69.11 | 42.87 | 64.65 | 68.18 | 31.82 | 67.81 |
| COLES-GCN | Both | 60.74 | 45.49 | 59.33 | 63.28 | 37.54 | 59.17 | 63.46 | 25.73 | 63.42 |
| COLES-GCN (Stiefel) | Both | 62.46 | 47.01 | 59.38 | 65.17 | 38.90 | 60.85 | 63.56 | 25.81 | 63.58 |
| COLES-SGC | Both | 65.62 | 52.32 | 56.95 | 68.24 | 43.09 | 63.85 | **69.47** | 32.31 | **68.57** |
| COLES-S$^2$GC | Both | **69.70** | **55.35** | 63.06 | **69.20** | **44.41** | **64.70** | 68.76 | **33.42** | 68.12 |

## 6.4 Node Clustering

We compare COLES-GCN and COLES-S$^2$GC with three types of clustering methods listed below:

i. Methods that use only node features *e.g.*, k-means and spectral clustering (spectral-f) construct a similarity matrix with the node features by a linear kernel.

ii. Structural clustering methods that only use the graph structure: spectral clustering (spectral-g) that takes the graph adjacency matrix as the similarity matrix, and DeepWalk [35].

iii. Attributed graph clustering methods that use node features and the graph: Graph Autoencoder (GAE), Graph Variational Autoencoder (VGAE) [22], Adversarially Regularized Graph Autoencoder (ARGE), Variational Graph Autoencoder (ARVGE) [34], SGC [49] and S$^2$GC [61].

We measure the performance by the clustering Accuracy (Acc), Normalized Mutual Information (NMI) and macro F1-score (F1). We run each method 10 times on Cora, CiteSeer and PubMed. We report the clustering results in Table 8. We set the number of propagation steps to 8 for SGC, S$^2$GC, COLES-SGC and COLES-S$^2$GC, following the setting of [57]. We note that COLES-S$^2$GC outperforms S$^2$GC in most cases, whereas COLES-GCN outperforms contrastive GCN on all datasets.

**Scalability.** GraphSAGE and DGI require neighbor sampling which result in redundant forward/backward propagation steps (long runtime). In contrast, COLES-S$^2$GC enjoys a straightforward implementation which reduces the memory usage and runtime significantly. For graphs with more than 100 thousands nodes and 10 millions edges (Reddit), our model runs smoothly on NVIDIA 1080 GPU. Even on larger graph datasets, the closed-form solution is attractive as for COLES-S$^2$GC, the cost of eigen-decomposition depends on $d$ (a few of seconds on Reddit). The runtime of COLES-S$^2$GC is also favourable in comparison to multi-view augmentation-based GraphCL. Specifically, COLES-S$^2$GC took 0.3s, 1.4s, 7.3s and 16.4s on Cora, Citeseer, Pubmed and Cora Full, respectively. GraphCL took 110.19s, 101.0s, $\geq$ 8h and $\geq$ 8h respectively.

## 7 Conclusions

We have proposed a new network embedding, COnstrative Laplacian EigenmapS (COLES), which recognizes the importance of negative sample pairs in Laplacian Eignemaps. Our COLES works well with many backbones, *e.g.*, COLES with GCN, SGC and S$^2$GC backbones outperforms many unsupervised, contrastive and (semi-)supervised methods. By applying the GAN-inspired analysis, we have shown that SampledNCE with the sigmoid non-linearity yields the JS divergence. However, COLES uses the RBF non-linearity, which results in the Kantorovich-Rubinstein duality; COLES essentially minimizes a surrogate of Wasserstein distance, which offers a reasonable transportation plan, and helps avoid pitfalls of the JS divergence. Moreover, COLES takes advantage of the so-called block-contrastive loss whose family is known to perform better than their pair-wise contrastive counterparts. Cast as the alignment and uniformity losses, COLES enjoys the more robust geometric mean rather than the arithmetic mean (used by SoftMax-Contrastive) as the uniformity loss.

## Acknowledgments and Disclosure of Funding

We would like to thank the reviewers for stimulating questions that helped us improve several aspects of our analysis. Hao Zhu is supported by an Australian Government Research Training Program (RTP) Scholarship. Ke Sun and Piotr Koniusz are supported by CSIRO's Machine Learning and Artificial Intelligence Future Science Platform (MLAI FSP).

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
