# Contrastive Laplacian Eigenmaps
# (Supplementary Material)

**Hao Zhu**[†,§]  **Ke Sun**[§,†]  **Piotr Koniusz**[∗,§,†]
[§]Data61/CSIRO  [†]Australian National University
allenhaozhu@gmail.com, sunk@ieee.org, piotr.koniusz@data61.csiro.au

## A  Derivations of Contrastive Laplacian Eigenmaps

In this section, we perform the transition of Eq. (4) into Eq.(5). We note that Eq. (4) relies on two terms: $\mathbb{E}_{v \sim p_d(v)}\left[\mathbb{E}_{u \sim p_d(u|v)}(\mathbf{u}^\top \mathbf{v})\right]$ and $\eta \mathbb{E}_{v \sim p_d(v)}\left[\mathbb{E}_{u' \sim p_n(u'|v)}\left(-\mathbf{u'}^\top \mathbf{v}\right)\right]$. The above two terms are evaluated over two different distributions $u \sim p_d(u \mid v)$ and $u' \sim p_n(u' \mid v)$, respectively. Below we discuss how to reformulate $\mathbb{E}_{v \sim p_d(v)}[\mathbb{E}_{u \sim p_d(u|v)}\mathbf{u}^\top \mathbf{v}]$ into the matrix form $\mathbf{Y}^\top \mathbf{L}\mathbf{Y}$ (reformulation of the second term can be performed by analogy), where $\mathbf{L} = \mathbf{I} - \mathbf{W}^{(+)}$ and $\mathbf{W}^{(+)} = \mathbf{D}^{-1/2}\widehat{\mathbf{W}}\mathbf{D}^{-1/2}$ Let $p_d(v) = \frac{1}{\sqrt{D_{vv}}}$ and $p_d(u \mid v) = \frac{W_{uv}}{\sqrt{D_{uu}}}$. Then we have:

$$\mathbb{E}_{v \sim p_d(v)}[\mathbb{E}_{u \sim p_d(u|v)}\mathbf{u}^\top \mathbf{v}] = \sum_{u,v} \frac{\widehat{W}_{uv}}{\sqrt{D_{vv}D_{uu}}}\mathbf{u}^\top \mathbf{v} = \sum_i^{d'} \sum_{u,v} \frac{\widehat{W}_{uv}}{\sqrt{D_{vv}D_{uu}}}u_i v_i. \qquad (15)$$

Note our slight abuse of notations where $u_i$ and $v_i$ are $i$-th coefficients of vectors $\mathbf{u}$ and $\mathbf{v}$, whereas $u$ and $v$ are note indexes. We notice that $\sum_{u,v} \frac{\widehat{W}_{uv}}{\sqrt{D_{vv}D_{uu}}}u_i v_i$ has a bilinear form $\langle \mathbf{Y}_i, \mathbf{W}^{(+)}\mathbf{Y}_i \rangle$, which leads to:

$$\sum_i^{d'} \mathbf{y}_i^\top \mathbf{W}^{(+)}\mathbf{y}_i = \text{Tr}(\mathbf{Y}^\top \mathbf{W}^{(+)}\mathbf{Y}), \qquad (16)$$

where $\mathbf{u}$ and $\mathbf{v}$ are rows of $\mathbf{Y}$. Moreover, $u_i$ denotes the $i$-th element of the vector $u$ and $\mathbf{y}_i$ is the $i$-th column of the matrix $\mathbf{Y}$.

By analogy, if we sample $\kappa$ for the random graph (negative graph), we have:

$$\mathbb{E}_{v \sim p_d(v)}[\mathbb{E}_{u' \sim p_n(u'|v)} - \mathbf{u'}^\top \mathbf{v}] = -\frac{1}{\kappa}\sum_{k=1}^\kappa \mathbb{E}_{v \sim p_d(v)}[\mathbb{E}_{u' \sim p_n^*(u'|v)}\mathbf{u'}^\top \mathbf{v}] = -\frac{1}{\kappa}\sum_{k=1}^\kappa \text{Tr}(\mathbf{Y}^\top \mathbf{W}_k^{(-)}\mathbf{Y}),$$
$$\qquad (17)$$

where $p_n^*(u'|v)$ represents some uniform probability $p' > 0$ of creating the negative links between nodes $u'$ and $v$, which results in a sparse matrix $\mathbf{W}_k^{(-)}$. Averaging $\kappa$ times over such adjacency matrices is equivalent to sampling from the negative distribution $p_n(u'|v)$.

Combining Eq. (16) and (17) gives Eq. (5).

**Block-Contrastive Loss.**  Based on Eq. (15), we can extend Eq. (11) into two different items:

$$\mathbb{E}_{u \sim p_d(u|v)}(\mathbf{u}^\top \mathbf{v}) = \mathbf{v}^\top \sum_u \frac{\widehat{W}_{uv}}{\sqrt{D_{vv}D_{uu}}}\mathbf{u} = \mathbf{v}^\top \sum_u W_{uv}^{(+)}\mathbf{u}, \qquad (18)$$

and

$$\mathbb{E}_{u' \sim p_n(u'|v)}\left(\mathbf{u'}^\top \mathbf{v}\right) = \mathbf{v}^\top \sum_{u'} W_{u'v}^{(-)}\mathbf{u'}. \qquad (19)$$

---

[∗]The corresponding author.  Code: `https://github.com/allenhaozhu/COLES`.

35th Conference on Neural Information Processing Systems (NeurIPS 2021).

Thus, we have $\boldsymbol{\mu}^+ = \sum_u W_{uv}^{(+)}\mathbf{u}$ and $\boldsymbol{\mu}^- = \sum_{u'} W_{u'v}^{(-)}\mathbf{u}'$ in our case. For brevity, we omit $b$ in the above result, whose role in Eq. (11) is to normalize by the block size $e.g.$, the number of links between $v$ and $u$ (and some $b'$ for $v$ and $u'$, respectively). Based on the above derivations, Eq. (11) can be reformulated as:

$$-\mathbb{E}_{u\sim p_d(u|v)}(\mathbf{u}^\top\mathbf{v})+\mathbb{E}_{u'\sim p_n(u'|v)}\left(-\mathbf{u}'^\top\mathbf{v}\right) = -\mathbf{v}^\top(\boldsymbol{\mu}^+ - \boldsymbol{\mu}^-) = -\mathbf{v}^\top\sum_u(W_{uv}^{(+)} - W_{uv}^{(-)})\mathbf{u}. \quad (20)$$

Thus, $\sum_{u,v}(W_{uv}^{(+)} - W_{uv}^{(-)})\mathbf{u}^\top\mathbf{v}$ with the corresponding matrix form $\mathrm{Tr}(\mathbf{Y}^\top(W^{(+)} - W^{(-)})\mathbf{Y})$.

## B  Graph Homophily Predicts that COLES Outperforms SampledNCE with Sigmoid (an Intuitive Illustration)

Let us define the graph homophily for graph $G^{(+)}$ with the degree-normalized adjacency matrix $\mathbf{W}^{(+)}$, $n$ nodes and multiclass labels $l_1, \cdots, l_n$ as:

$$\mathcal{H}\big(G^{(+)}\big) = \frac{1}{n}\sum_{i=1}^n \frac{1}{|\mathcal{N}_i|}\sum_{j\in\mathcal{N}_i}\delta(l_i - l_j) = \frac{1}{n}\sum_{i=1}^n\sum_{j=1}^n W_{ij}^{(+)}\delta(l_i - l_j), \quad (21)$$

where $\delta(l_i - l_j)$ equals one if $l_i$ equals $l_j$, zero otherwise.

Furthermore, for negative sampling, we use the so-called negative graph $G^{(-)}$, which is a sparse graph with the uniform probability $p' > 0$ of connection between each pair of nodes. Thus, in expectation, the homophily of this graph is equal to homophily for the fully-connected graph, and is given by:

$$\mathcal{H}\big(G^{(-)}\big) = \frac{1}{n}\sum_{i=1}^n\sum_{j=1}^n W_{ij}^{(-)}\delta(l_i - l_j) = \frac{1}{C}\sum_{c=1}^C \rho_c^2, \quad (22)$$

where $C$ is the number of classes, $\rho_1, \cdots, \rho_C$ are class probabilities $e.g.$, $\rho_1 = 0.1$ means that class one is given to the 10% of nodes.

Looking at Eq. (8), we notice that for the SampledNCE with sigmoid, one can think of $D(x)$ and $1 - D(x')$ as a sigmoid for $xp_r$ and a reverse sigmoid for $p_g$, respectively. Therefore, to understand how well two distributions are separated, one can measure:

$$\mathrm{Sep}(v, u, u') = \frac{|D(\mathbf{u}^\top\mathbf{v}) - D(\mathbf{u}'^\top\mathbf{v})|}{D(\mathbf{u}^\top\mathbf{v}) + D(\mathbf{u}'^\top\mathbf{v})}, \quad (23)$$

where $\mathrm{Sep}(v, u, u') \to 1$ if the dot-products of embeddings $\langle\mathbf{v}, \mathbf{u}\rangle$ and $\langle\mathbf{v}, \mathbf{u}'\rangle$ can be separated from each other linearly, and $Sep(v, u, u') \to 0$ if they cannot be separated.

To this end, we make a simple assumption. If $\mathcal{H}\big(G^{(+)}\big) \to \mathcal{H}\big(G^{(-)}\big)$, this means that $p_r$ and $p_g$ become highly similar, which is good for the underlying JS divergence but it means that is impossible to find embeddings which will separate two distributions (contrastive learning fails in this regime). At the other extreme end, $\mathcal{H}\big(G^{(+)}\big) \gg \mathcal{H}\big(G^{(-)}\big)$, which indicates that we can easily find embeddings that separate $p_r$ and $p_g$. However, these embeddings can be disjoint, which is manageable for the underlying surrogate of Wasserstein distance ins COLES but is hard for SampledNCE with sigmoid with the underlying JS divergence.

To validate our intuition, Figure 2 shows $\Delta\mathrm{Acc.}$ between COLES-GCN and GCN+SampledNCE as a function of $\Delta\mathcal{H} = \mathcal{H}\big(G^{(+)}\big) - \mathcal{H}\big(G^{(-)}\big)$. We use the same experimental setting as the one used for results reported in Table 2. After sorting results by homophily in the ascending order, we note that the overall trend agrees with our expectations that for small $\Delta\mathcal{H}$, both methods struggle more as it is harder for the contrastive setting to find distinctive embeddings. However, as $\Delta\mathcal{H}$ increases, the overlap between $p_r$ and $p_g$ decreases, making it easier to find distinctive embeddings. COLES benefits a lot under this setting, whereas SampledNCE with sigmoid benefits to a lesser degree.

The above simple illustration/intuition is by no means an exhaustive proof given we evaluated it on only three datasets, and embeddings can exploit often complex neighborhood patterns which the homophily index cannot capture (something appearing as random from the homophily perspective may still enjoy an informative complex pattern). Nonetheless, our observation supports our claim that COLES works well in the regime where contrastive learning is easily viable, whereas the SampledNCE with sigmoid struggles more by contrast.

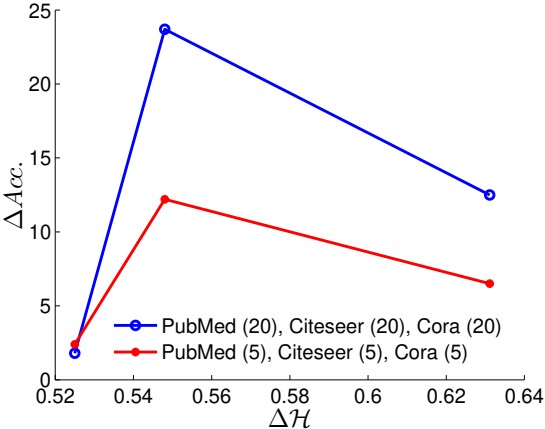

Figure 2: $\Delta$Acc. between COLES-GCN and GCN+SampledNCE w.r.t. $\Delta\mathcal{H}$. We sorted the results by $\Delta\mathcal{H}$. Thus, the first, second and third point on each curve (from left to right) corresponds to Pubmed, Citeseer and Cora, respectively. In parentheses, we indicate the number of labeled training samples per class.

# C Reproducibility

## C.1 Datasets

In this paper we use six datasets to evaluate our method. Cora is a well-known citation network labeled according to the paper topic. Most of approaches report on a small subset of this dataset. The **Cora** dataset consists of 2708 scientific publications classified into one of seven classes. The citation network consists of 5429 links. Each publication in the dataset is described by a 0/1-valued word vector indicating the absence/presence of the corresponding word from the dictionary. The dictionary consists of 1433 unique words. **Cora Full** consists of 19793 scientific publications classified into one of seventy classes. The citation network consists of 65311 links. The dictionary consists of 1433 unique words.

The **CiteSeer** dataset consists of 3312 scientific publications classified into one of six classes. The citation network consists of 4732 links. Each publication in the dataset is described by a 0/1-valued word vector indicating the absence/presence of the corresponding word from the dictionary. The dictionary consists of 3703 unique words.

The **Pubmed** dataset consists of 19717 scientific publications from PubMed database pertaining to diabetes classified into one of three classes. The citation network consists of 44338 links. Each publication in the dataset is described by a TF/IDF weighted word vector from a dictionary which consists of 500 unique words.

The **Reddit** dataset is a graph dataset from Reddit posts made in the month of September, 2014. The node label in this case is the community, or subreddit, that a post belongs to. The 50 large communities have been sampled to build a post-to-post graph, connecting posts if the same user comments on both. In total, this dataset contains 232,965 posts with an average degree of 492. The first 20 days are used for training and the remaining days for testing (with 30% used for validation).

The **Ogbn-arxiv** dataset contains a directed graph, representing the citation network between all Computer Science (CS) arXiv papers indexed by MAG [46]. Each node is an arXiv paper and each directed edge indicates that one paper cites another one. Each paper comes with a 128-dimensional feature vector obtained by averaging the embeddings of words in its title and abstract. The embeddings of individual words are computed by running the skip-gram model [33] over the MAG corpus. We also provide the mapping from MAG paper IDs into the raw texts of titles and abstracts here. In addition, all papers are also associated with the year that the corresponding paper was published.

Table 9: The hyperparameters of datasets (node classification).

| Dataset | Optimizer | K | lr | weight decay | Epoch | hidden size |
|---------|-----------|---|------|-------------|-------|-------------|
| Cora | Adam | 8 | 1e-3 | 5e-4 | 20 | 512 |
| Citeseer | Adam | 8 | 1e-4 | 1e-4 | 80 | 512 |
| Pubmed | Adam | 8 | 2e-2 | 1e-5 | 40 | 256 |
| Cora Full | Adam | 2 | 1e-2 | 0 | 30 | 512 |
| Ogbn-arxiv | SVD | 10 | None | None | 500 | 126 |
| Reddit | SVD | 2 | None | None | None | 600 |

Table 10: The hyperparameters of datasets (node clustering).

| Dataset | Optimizer | K | lr | weight decay | Epoch | hidden |
|---------|-----------|---|------|-------------|-------|--------|
| Cora | Adam | 8 | 1e-2 | 5e-4 | 1 | 512 |
| Citeseer | Adam | 8 | 1e-4 | 1e-4 | 30 | 512 |
| Pubmed | Adam | 8 | 2e-2 | 1e-5 | 40 | 256 |

# D   Implementation

We use PyTorch to implement COLES and its variants. The propagation procedure is efficiently implemented with sparse-dense matrix multiplications. The codes of GCN, COLES-GCN, SGC, COLES-SGC, S$^2$GC and COLES-S$^2$GC are also implemented with PyTorch. The weight matrices of classifier are initialized with Glorot normal initializer. We employ Adam [21] to optimize parameters of the proposed methods and adopt early stopping to control the training epochs based on validation loss. For the experiments on Cora, Citeseer, Pubmed, CoraFull, we use SGD to optimize Eq. (5) because the datasets are small enough. For reddits and Ogbn-arxiv, we use Eq. (6) to obtain the closed-form solution to accelerate the speed. All the experiments in this paper are conducted on a single NVIDIA GeForce RTX 1080 with 8 GB memory. Server operating system is Unbuntu 18.04. As for software versions, we use Python 3.7.3, PyTorch 1.6.0, NumPy 1.18.1, SciPy 1.4.1, CUDA 9.1.

## D.1   Hyperparameters

We did not put much effort to tune these hyperparameters in practice, as we observe that COLES is not very sensitive to different hyperparameters. SGC and S$^2$GC use the aggregation step $K$, the only hyperparameter for these methods. Thus we use $K = 8$ for most benchmarks. Except for Ogbn-arxiv, we use logistic regression as the classifier for all contrastive based methods. Note that we do not tune any parameter for the logistic regression and just use the default setting. In Ogbn-arxiv, the given features are non-linear because they are based on Bag-of-Words with word embeddings. Thus, the MLP classifier is selected for COLES-S$^2$GC. Specifically, we keep the setting of the MLP classifier in the baseline. There are two hidden state layers, and the hidden state size is 256 dimension for each layer. The learning rate for the MLP is 0.005 and the dropout rate is 0.4.