# OpenReview forum: "Contrastive Laplacian Eigenmaps"
_NeurIPS.cc/2021/Conference — NeurIPS 2021 Poster_

### Official Review · Reviewer_8s4v · 2021-07-08

**Rating:** 6
**Confidence:** 3

**Summary:**

In this work, the authors improved contrastive learning of graph node embedding from the viewpoint of Laplacian eigenmap. Removing nonlinear activation used in traditional NCE, the authors reformulate NCE as a randomized Laplacian eigenmap problem, which can be used to learn the GNNs with linear architectures. The authors claimed that the proposed method is equivalent to minimize the expectation of the Wasserstein distance between the positive and the negative distributions conditioned on the target node.

-- after rebuttal ---
Most of my concerns are solved, so I change my score from 5 to 6.

**Limitations And Societal Impact:**

I did not find any obvious or potential negative societal impact of this work.

**Main Review:**

Pros:
1. The paper is written clearly.

2. The experimental results of different tasks demonstrate the usefulness of the proposed method. The implementation details are provided.

3. Connecting NCE with Laplacian eigenmap provides an interesting paradigm of node embedding, which is promising.

Cons:
1. The introduction of GAN between Line 121 and Line 137 is redundant in my opinion — the conclusion has been given in the paper of WGAN. Compressing this part will give more space and the authors can move more derivations from the supplementary file to the main paper.

2. The Lipschitz continuity of g(u) (and COLES) becomes obvious because the authors have claimed the linearity of the target model. Additionally, it seems unnecessary to derive “||u-u’||_2” in Line 155 and the following inequality.

3. I am not sure about the superiority of “block-contrastive loss”. As the authors claimed in 163, the block-contrastive loss is a lower bound of pairwise-contrastive loss. However, if my understanding is correct, the objective function is minimizing the loss. I don’t think minimizing a lower bound of an objective function. (For example, EM is reasonable because it minimized an upper bound of MLE; a main drawback of WGAN is that it can just minimize a lower bound of Wasserstein distance.)

4. Additionally, different from classic WGAN, the authors consider the Wasserstein distance between conditional generation and the condition corresponds to the model parameters (i.e. v). In other words, learning the embedding u’s conditioned on v will influence the learning of other nodes’ embeddings that conditioned on the u’s. Such a non-i.i.d. situation makes the feasible domain of g_v(u) in Eq. (11) changes during training. Maybe the authors should emphasize and analyze this difference.

5. The parameters of COLES are constrained (as shown in Eq. (6)). How to achieve the constraints in the framework of SGD is not explained clearly.


**Time Spent Reviewing:**

2

---

> ### Author Response · Authors · 2021-08-10
> **Response to Rev. 4 (8s4v)**
>
> We thank the reviewer for providing detailed comments and questions that help shape and improve our work. Below we respond to each comment in hope to clarify some misunderstandings.
>
> ## 1. The introduction of GAN in line 121-137  redundant in my opinion.
>
> Thank you. We agree and we have removed now  Eq. 8 and 9 (prior knowledge) and compressing the surrounding text. **We just wanted to politely highlight that the idea of analysing SampledNCE by plugging it into the WGAN framework is nonetheless our contribution** explaining why COLES performs well. Thus, we focused now more on not how the JS divergence emerges in the WGAN but how COLES turns into the the Kantorovich-Rubinstein duality.
>
> \subsection{The Lipschitz continuity of g(u) (and COLES) becomes obvious because the authors have claimed the linearity of the target model.}
>
> Thank you, indeed, $g\_v(\mathbf{u})$ can be seen as Lipschitz continuous and we can move this to the Suppl. Material. We provided the equation between lines 155--156 because **we wanted to show that with three variables ($\mathbf{u}$, $\mathbf{v}$ and $\mathbf{u}'$), the entire expression $\vert \mathbf{u}^\top\mathbf{v}-\mathbf{u}'^\top\mathbf{v}\vert$  is  Lipschitz continuous (consider the dot-product)** with the $K'=\Vert \mathbf{v} \Vert\_{\max}$ constant in $\vert \mathbf{u}^\top\mathbf{v}-\mathbf{u}'^\top\mathbf{v}\vert\leq K'\Vert\mathbf{u} - \mathbf{u}' \Vert_1$.
>
> ## 2. It seems unnecessary to derive $||\mathbf{u}-\mathbf{u}’||_2$ in Line 155.
>
> Duly noted, we have removed this.
>
> ## 3. I am not sure about the superiority of block-contrastive loss.
>
> We believe there is a misunderstanding due to the misuse of language. **Saunshi et al. [27] have shown that block-contrastive losses attain the lower objective compared to their equivalent of element-wise contrastive losses**. Essentially, they have argued that the lower objective of block-contrastive losses results in their better generalization capability compared to the element-wise equivalent (which happens to be also an upper bound of the block-contrastive).
>
> In our manuscript, **we do not mean that we derive and optimize a lower bound of the COLES loss**. What we mean is that the COLES loss has a form of block-contrastive loss.
>
> We optimize COLES, not some lower bound of it (and we apologise for the confusion). We understand that for GMMs, the objective is intractable, thus a tractable upper bound is devised and minimized.
>
> We have fixed now this `misused terminology' issue.
>
>
> ## 4. Different from classic WGAN, the authors consider the Wasserstein distance between conditional generation and the condition corresponds to the model parameters (i.e. v). In other words, learning the embedding u’s conditioned on v will influence the learning of other nodes’ embeddings that conditioned on the u’s. Such a non-i.i.d. situation makes the feasible domain of $g\_v(u)$ in Eq. (11) change during training. Maybe the authors should  analyze this difference.
>
>
>
> By Eq. 11, **our objective function is a lower bound of the supremum in the dual formulation of the EMD**. Therefore our loss (opposite of objective) is a surrogate function of the corresponding minimization problem. We are essentially minimizing the EMD as well as the bound gap. This is similar to the EM approach.
>
> Globally, COLES is structured into two layers: the graph neural network to obtain the graph embedding vectors, followed by the contrastive learning module to compute the loss. The node embeddings are jointly learned in this framework. **The fact that the anchor node $v$ contains free parameters allows the $g_{\mathbf{\Theta}}$ function to explore a subset of functions which satisfies the Lipschitz condition, so that the bound gap can be reduced**
> (if we can enumerate the set of all functions satisfying the Lipschitz condition,
> then the gap is reduced to zero.)
> **No matter $v$ is free or not, our loss is a surrogate of the Wasserstein distance.**
>
> Moreover,  nodes $u$ are sampled from the vicinity of node $v$ according to the underlying graph (imagine that $u$ and $v$ are in the same cluster). Thus, the mean of  embeddings $\mathbf{u}$ (sampled in the vicinity of node $v$) is brought closer to the anchor $\mathbf{v}$, and vice versa. **The choice of $\mathbf{v}$ within the cluster does not change the fact that all embeddings in the cluster are brought to be close to each other in the cosine sense, and the embeddings of distant nodes are pushed away from the cluster.**
>
> Kindly note that **the optimization case described by the reviewer is exactly the same as for SampledNCE.** However, our embeddings are $\ell_2$ norm normalized. The normalization to the sphere radius $\tau$ gives the non-changing domain of $-\tau^2\leq g\_v(\mathbf{u})\leq \tau^2$.
>
> ## 5. How to achieve the constraints in the framework of SGD?
>
>
> Kindly note that **our results and the learning algorithm are not affect by the exact constraints in Eq. 5**.
> **The purpose of the constraint is to prevent the embedding from blowing up and to satisfy the Lipschitz condition.** We have:
> $$f^\top\_{\mathbf\Theta}(\mathbf{X}) f\_{\mathbf\Theta}(\mathbf{X})
> =\sum\_{i}f\_{\mathbf\Theta}(\mathbf{X}\_i)^\top f\_{\mathbf\Theta}(\mathbf{X}\_i)\approx\sum\_{i\in\mathcal{B}}f\_{\mathbf\Theta}(\mathbf{X}\_i)^\top f\_{\mathbf\Theta}(\mathbf{X}\_i)$$
> where $\mathbf{X}\_i$ is the $i$'th input row vector,
> $\mathcal{B}$ is a random subset of samples,
> and the estimation is unbiased.
> Therefore, the constraint imposed on
> $f^\top\_{\mathbf\Theta}(\mathbf{X}) f\_{\mathbf\Theta}(\mathbf{X})$
> can be equivalently imposed on its minibatch estimation
> $\sum_{i\in\mathcal{B}}f^\top_{\mathbf\Theta}(\mathbf{X}\_i) f\_{\mathbf\Theta}(\mathbf{X}\_i)$. This follows the same principle
> as SGD.
>
> In our submission, **we tried two approaches to attain  the Lipschitz continuity**, that is, (i) we simply imposed a soft penalty $-\beta\lVert f^\top\_{\mathbf\Theta}(\{\mathbf{X}\_i\}\_{i\in\mathcal{B}}) f\_{\mathbf\Theta}(\{\mathbf{X}\_i\}\_{i\in\mathcal{B}}) -\mathbf{I}\rVert\_F^2$ and (ii) we enforced the $\ell\_2$ norm normalization of node embeddings.
>
>
>
> Moreover, **we have modified our Eq. 6 (main submission) to the following batch-based form:**
>
> For a graph network $f_\mathbf{\Theta}(\mathbf{X})$, we can write:
> $$\mathbf{P}^*,\mathbf{\Theta}^* = \arg\max\limits\_{\mathbf{P},\mathbf{\Theta}} \text{Tr}(\mathbf{P}f\_\mathbf{\Theta}(\mathbf{X})^\top\Delta\mathbf{W}f_\mathbf{\Theta}(\mathbf{X})\mathbf{P}^\top),\text{ s.t. }\mathbf{P}\mathbf{P}^\top=\mathbf{I}.$$
> and solve for $\mathbf{P}$ (wide matrix) on **the Stiefel manifold via the GeoTorch package** (`*Trivializations for gradient-based optimization on manifolds*' by Lezcano-Casado available at https://github.com/Lezcano/geotorch), which provides a number of constrained optimization subroutines solving problems on a chosen manifold.
>
>
> Below is the table summarizing the results for these three strategies:
>
> |                           | Cora (5)  | Cora (20) | Citeseer (5) | Citeseer (20) | Pubmed (5) | Pubmed (20) | Cora-full (5) | Cora-full (20) |
> |---------------------------|-----------|-----------|--------------|---------------|--------------|---------------|--------------|---------------|
> | COLES-GCN-L2Norm                     | 72.8±4.1  | 80.5±1.2  | 60.2±3.0     | 68.4±1.1      | 62.8±4.9     | 72.1±2.4      |42.7±1.7     | 55.8±0.6 |
> | COLES-GCN-Soft Orthogonal                     | 73.8±3.4  | 80.8±1.3  | 66.0±2.6     | 69.0±1.0      | 62.7±4.6     | 72.7±2.1      |47.3±1.5     | 58.9±0.5 |      |
> | **COLES-GCN-Stiefel Orthogonal**         | 75.0±3.4   | 81.0±1.3  | 67.9±2.3     | 71.7±0.9     |62.6±5.0     | 73.2±2.6     | 47.6±1.2     | 59.2±0.5         |
>
> As can be seen, using the constrained optimization on the Stiefel manifold, following our modified Eq. 6 **we improve   resutls further by up to 2.7\% in some cases**.
>
> We hope that our clarifications are sufficient to convince the reviewer that our work is solid and worth acceptance. We have genuinely put a lot of considerations into our theory and implementation of COLES. It achieeves state-of-the-art results not by mere engineering (like so many papers these days) but by a careful mathematical design of our contrastive setting.

---

> > ### Comment · Reviewer_8s4v · 2021-09-03
> > **After rebuttal**
> >
> > Thank the authors for their detailed explanations. I am satisfied with the response and would like to see these changes in the final version. Therefore, I change my score to 6.

---

> ### Comment · Area_Chair_XLe1 · 2021-08-25
> **Reviewer response required**
>
> Dear Reviewer 8s4v,
>
> The authors have provided a very thorough response to your review. Has any of this changed your mind? Would you be willing to raise your score in response to their comments?

---

### Official Review · Reviewer_mr5C · 2021-07-12

**Rating:** 7
**Confidence:** 4

**Summary:**

This paper provides a reformulation of the traditional Laplacian Eigenmaps in a contrastive setting, for graph nodes embedding. Theoretical results for its connection to Wasserstein distance and block-contrastive loss are presented. Overall, I think this is a good paper with substantial contributions in connecting manifold learning and constructive learning.

**Limitations And Societal Impact:**

I understand this paper focuses more on methodology part, but I suggest the authors should discuss the potential negative societal impact of their work,  probably in terms of applications. It is currentluy missing from the manuscript.

**Main Review:**

Pros:
      1) This paper is organised well and clearly written.
      2) The idea to connect the Laplacian Eigenmaps and contrastive learning is novel.
      3) There are both theoretical analysis and empirical results supporting the proposed method.


Cons:
      1) It would be better if the authors can give more details on how the adjacency $\textbf{ W}$ is generated.
If the $\textbf{ W}$ is generated through k-nearest neighbours how can the computational complexity be handled reasonably with large datasets?
      2) The examples on when the methods using JS divergence may underperform have been given, but I think the paper would also benefit from providing more insights on when the proposed method would outperform the traditional Laplacian Eignemaps.

**Time Spent Reviewing:**

3.5

---

> ### Author Response · Authors · 2021-08-10
> **Response to Rev. 3 (mr5C)**
>
>
> We thank the reviewer for the insightful questions, especially on the performance pros and cons of the JS divergence and the COLES.
>
> ## 1. How is the adjacency matrix $\mathbf{W}$ generated?
>
>
> In our experiments, **to filter out the impact of graph construction methods on COLES, we directly use standard datasets such as CORA, Citeseer, Pubmed which provide their own relational graphs with binary links** (standard practice). These datasets  contain the adjacency matrix and corresponding node attributes. **The only step** we need to perform to obtain $\mathbf{W}$ is to take an unnormalized adjacency matrix $\mathbf{\widehat{W}}$ and **normalize (divide) it by the square root of its degree matrix**:
> $$\mathbf{W}=\mathbf{D}^{-1/2}\mathbf{\widehat{W}}\mathbf{D}^{-1/2}$$
> **This can be readily obtained with the spare-sparse matrix multiplication** (pytorch-geometric and pytorch-sparse provide support).
>
> For generic problems, there are many ways to generate adjacency matrix efficiently by: (i) **Fast Approximate Nearest Neighbour (ANN)** search that returns approximate top-k neighbours, (ii) **RBF function to obtain some similarity measure** which can then be truncated if sparsity is required or, (iii) by the compact support kernels such as the triangular kernel.
>
> **To obtain the negative $\textbf{W}^{(-)}$, we proceed as follows**. Given a node, we randomly draw edges (i.i.d.) for other k nodes according to the uniform distribution. Subsequently, we obtain a random matrix $\widetilde{\mathbf{W}}$. To ensure the random matrix is symmetric (semi-)definite positive, we have $\textbf{W}^{(-)}=\frac{1}{2}(\widetilde{\mathbf{W}}+\widetilde{\mathbf{W}}^\top)$ which captures an undirected random graph.
>
> ## 2. Insights about when JS divergence is better than the  Laplacian Eignemaps.
>
> Thank you. This is a very interesting question. As Figure 1 (main submission) indicates, **the JS divergence performs very well when** the positive and negative distributions of $\langle\mathbf{u},\mathbf{v}\rangle$ and $\langle\mathbf{u}',\mathbf{v}\rangle$ overlap heavily. If the underlying learner (backbone) cannot separate these distributions, this indicates that the contrastive learning cannot be achieved e.g., a better representation of neighborhood is needed or simply the class/feature distribution of node neighbours and the class/feature distribution of random graphs are on average identical. Either way, we expect that JS performs well in such situations. However, its performance deteriorates as the contrastive learning progresses in separating the positive pairs from negative pairs.
>
> **In the Suppl. Material we have provided an analysis which suggests that the larger is the Graph Homophily index $\Delta\mathcal{H}$, the better are results of COLES** (Figure 2, Suppl. Mat.) This suggests that **COLES copes well under poor overlap of positive and negative distributions** (due to the Wasserstein property) because the low Graph Homophily indicates that neighborhoods contain nodes of mixed class labels (thus mixed embedding features). For that reason, the distributions of labels in the graph neighbourhood and in the random graph may be similar (overlap). For the large Graph Homophily, nodes in the graph neighborhood have similar/same labels (and thus similar/same node embeddings), whereas a random graph should still exhibit a mixture of labels. For that reason, the positive and negative distributions overlap less, and thus they are more suited for comparison via the Wasserstein distance. We expect that **JS may perform better when the Graph Homophily is very low** but this is an unlikely scenario (most graphs have $\Delta\mathcal{H}>0.5$).
>
> Finally, Tables 3 and 4 (main submission) show the performance for Laplacian Eigenmaps ($\kappa=0$) and the Contrastive Laplacian Eigenmaps ($\kappa>0$). As is clear, Laplacian Eigenmaps suffer a lot. **Laplacian Eigenmaps  would perhaps work similarly to COLES if the Homophily index of a graph was very low** (hard to separate positive and negative distributions so contrastive learning may be not useful).
>
> ## 3. The potential negative societal impact of your work.
>
> **Like every tool made by humans, GNNs can be used for noble or evil purposes**. To paraphrase, a knife may cut bread or kill, a scalpel may save lives in a theatre.
>
> We believe COLES has no `special' negative impact compared with other GNNs. In fact, by yielding better results in semi-supervised setting, COLES shows better generalisation capacity which may result in hopefully one or two classification outcomes that are less biased or mistaken. Though, of course, fairness is a playground best left to experts working in that field, but their tools are applicable to our work.
>
> Of course, GNNs can be used for instance for inference of user behaviors in social networks (authorized or unauthorized) and crunching dependencies in big data graphs. Like all AI, the use of GNNs needs a regualtory framework regarding privacy, fairness, openness and trustworthiness but perhaps more important is the aspect of regulatory laws for storage of the data (GNNs are just algorithms, they cannnot do much of good or bad without the data). GNNs can be the force for good if appleid to smart cities, dealign with large metropolitan areas, traffic, energy and digital traffic planning, to give few examples. They can be useful in inferring sickness from health records etc. They help innovate in digital trading.
>
> We would like to highlight one important positive societal effect our network has. S$^2$GC+COLES (Eq. 6 for linear networks) is $\geq$100x faster than for instance augmentation-based contrastive pipelines (kindly see Resp. 3 to Rev. 2 (Gh8g)) and $\geq$100x faster than GRACE and GCA (recent contrastive pipelines (GRACE is cited in the main paper, GCA is provided in the Resp. 3 to Rev. 2) on the CORA dataset, which means **$\geq$100x energy saving**. Thus, **S$^2$GC+COLES has a small environmental footprint**. If one considers that COLES trains without labels, and the it can be adapted on semi-annotated datasets, that represents a further cost and energy savings compared to standard non-contrastive GCNs by avoiding the need for the full annotation of datasets (which requires energy expenditures too).

---

### Official Review · Reviewer_Gh8g · 2021-07-16

**Rating:** 5
**Confidence:** 5

**Summary:**

In this article, the authors extended the classic Laplacian eigenmaps to the form of contrastive learning and proposed a new graph contrastive representation learning model. The authors also gave some theoretical analysis about their proposed contrastive learning loss.

**Limitations And Societal Impact:**

Not sufficient

**Main Review:**

The authors extended the classic Laplacian eigenmaps to the form of contrastive learning, it seems that the experimental results are good. However, the problem studied in this paper is not new. I think the novelty is incremental. In the current version, I have the following suggestions:
1.	Eq. (6) should be explained in more detail about F.
2.	Missing some important related works about graph contrastive representation learning, e.g., Graph contrastive learning with augmentations (You et al., NeurIPS 2020).
3.	What are the advantages of the proposed method compared to You et al., NeurIPS 2020.
4.	The experimental results in Table 1 and 7 are not solid enough to verify the effectiveness of the proposed method, since there lack some state-of-the-art methods for comparison.


**Time Spent Reviewing:**

About eight hours.

---

> ### Author Response · Authors · 2021-08-10
> **Response to Rev. 2 (Gh8g)**
>
>
> We thank the reviewer for the valuable comments and pointing to us a very interesting related work. Below we answer all questions.
>
> ## 1. More detail about F in Eq. (6).
>
> Kindly note that we have given examples of $\mathbf{F}$ in **lines 112--114** but we will of course expand on these. By-and-large, $\mathbf{F}$ can be understood as **a filtering operator which captures the neighborhood of each node according to a specific design**. For SGC, $\mathbf{F}$ is simply defined as a diffusion of the graph adjacency matrix. For S$^2$GC, $\mathbf{F}$ is defined as the weighted average of consecutive degrees of diffusion of the graph adjacency matrix, which results in consecutively increasing receptive fields around each node. We have now added further details including an illustration in our draft.
>
> ## 2. The problem studied is not new.
>
> Thank you. We politely disagree and explain our contributions below. We hope that the kind reviewer will sympathise with us that our theoretical contributions are non-trivial and results are very promising for such a simple and fast pipeline. **Instead of  taking SampledNCE framework at its face value like many works on contrastive learning do, we show how to achieve a better contrastive setting**:
>
>
> + Contrastive learning, as a sub-field of machine learning, is not new but its topic is far from resolved as noted by many contributors of CVPR'21 tutorial on self-supervised learning: https://gidariss.github.io/self-supervised-learning-cvpr2021
>
> + Kindly note cited by us many contrastive GNN pipelines such as DeepWalk, Graph2Gauss, DGI, GRACE, GraphSAGE and GCN with SampledNCE mechanisms, which focus on the contrastive learning theory rather than augmentation strategies. Our submission also focuses on the theory and shows how to improve contrastive learning.
>
> + Our work is not incremental--**we are the first to cast the SampledNCE framework as the WGAN  framework** (lines 128--131 and the following text) which led us to **our novel Contrastive Laplacian Eigenmaps.**
>
> + We have provided a necessary **proof that COLES enjoys the Kantorovich-Rubinstein duality for the Wasserstein distance due to the Lipschitz continuity of COLES** w.r.t. $\mathbf{u}$, $\mathbf{u'}$ and $\mathbf{v}$ variables (Eq. 11), to prove the superiority of our loss over the traditional SampledNCE. We proposed the objective for SGD-based and linear GNNs in Eq. 1 and 6. We have also proved the **block-contrastive nature of our COLES** in Eq. 12 (block-contrastive losses provide a better generalization to traditional pairwise losses (Saunshi et al. [27])).
>
> In conclusion, **we have not seen anybody making this nice connection between classic methods (the Laplacian Eigenmaps) and their contrastive reformulation and reparametrization**. We show that **COLES beats pipelines based on the standard contrastive SoftMax and SampledNCE paradigms**.
>
> ## 3. Missing `Graph Contrastive Learning with Augmentations' by You et al., NeurIPS 2020.
>
> Thank you for highlighting this nice work to our attention. We have now cited it accordingly and compared to our work. We would like to kindly point a few of details below:
>
> + **You et al. study graph classification not the node classification which we study** (there is no mention of node classification in their paper). If we understood correctly, they train the network with multiple graphs and their augmentations via the standard SoftMax contrastive loss e.g., known from GRACE (cited in our paper) and  `*Distributed Representations of Words and Phrases and their Compositionality*’ by Mikolov et al.
>
>
> + Work of You et al. belongs to a multi-view family of methods which are not our competitor--our work is complementary e.g.,  **You et al. focus on the augmentations, whereas we do not use augmentations at all**. We instead study the nature of the sampling objective.
>
> + Graph classification is very different from node classification, as it requires e.g., a robust global aggregator and the ability to deal with node permutations (graph isomorphism) to describe entire graphs.
>
> Nonetheless, **we noticed  in the code repository of You et al. that their very nice work (referred below as GraphCL) was adapted informally to node classification**, thus we provide comparisons (OOM means Out of Memory on our 8GB GPU):
>
> |                           | Cora (5)  | Cora (20) | Citeseer (5) | Citeseer (20) | Pubmed (5) | Pubmed (20) | Cora-full (5) | Cora-full (20) |
> |---------------------------|-----------|-----------|--------------|---------------|--------------|---------------|--------------|---------------|
> | **COLES-S$^2$GC**                     | 76.5±2.6  | 81.5±1.2  | 67.5±2.2     | 71.3±1.0      | 66.0±5.2  | 77.4±1.9  | 50.8±1.4     | 61.8±0.5      |
> | S$^2$GC                     | 71.4±4.4   | 81.3±1.2  | 60.3±4.0     | 69.5±1.2     |67.6±4.2   | 73.3±2.0  | 41.8±1.7     | 60.0±0.5     |
> | GraphCL                   | 72.6±4.2  | 78.3±1.7  | 65.6±3.0     | 71.1±0.8     | OOM  | OOM  | OOM     | OOM     |
> | GCA                   | 61.5±4.9  | 75.8±1.9  | 43.2±3.6     | 55.7±1.9     | OOM  | OOM  | OOM     | OOM     |
>
> The table shows that **COLES-S$^2$GC outperforms GraphCL by up to 4\% (CORA) and up to 2\% (Citeseer)**. This is understandable as our work is designed for node classification, theirs for the graph classification (**we expect that is where their results shine**).
>
> We show that **our method can benefit from augmentations** of You et al. below:
>
> |                           | Cora (5)  | Cora (20) | Citeseer (5) | Citeseer (20) | Pubmed (5) | Pubmed (20) | Cora-full (5) | Cora-full (20) |
> |---------------------------|-----------|-----------|--------------|---------------|--------------|---------------|--------------|---------------|
> | COLES-GCN                     | 73.8±3.4  | 80.8±1.3  | 66.0±2.6     | 69.0±1.0      | 62.7±4.6     | 72.7±2.1      |47.3±1.5     | 58.9±0.5 |      |
> | **COLES-GCN-Augmentation**                     | 75.3±3.3   | 81.0±1.3  | 66.7±2.3     | 69.8±1.3     |63.9±5.0     | 73.4±2.5     | 48.0±1.2     | 59.4±0.5     |
>
> We obtain the boost of up to 1.5\% by applying the augmentation strategy of You et al. for COLES with GCN.
>
> The table below also shows that **our method is 366x and 72x faster on CORA and Citeseer** due to its linear nature and a simple quadratic loss (OOT means >8 hours):
>
> |Time Cost (Seconds)                           | Cora  | Citeseer  |  Pubmed  | Cora-full  |
> |---------------------------|-----------|-----------|-----------|-----------|
> | **COLES-S$^2$GC**                      | 0.3     | 1.4      | 7.3     |   16.4             |
> | GraphCL                   | 110.19  | 101.0  |  OOT |  OOT
>
>
> ## 4. Advantages of COLES compared to You et al.
>
> Thank you. The two works are very different and  complementary in their nature, as described below:
>
> +  **COLES results in a quadratic problem** with orthogonality constraints. Thus, paired with SGC or S$^2$GC linear networks, **it enjoys a unique optimum** (no local minima) and no flat optimization regions (**no vanishing gradients**).
> + **COLES enjoys the WGAN formulation** (Eq. 11, main paper) which lets it cope well under a poor overlap of positive pair and negative pair distributions (Figure 1, main paper). In contrast, the SoftMax contrastive loss (Eq. 3,  You et al.) and its SampledNCE variant (Eq. 4, You et al.) do not enjoy these properties.
> + **COLES enjoys the block-contrastive loss** (Eq. 12, main submission). In contrast, the loss functions in You et al. do not exhibit such properties. The block-contrastive family of losses were shown to be superior in performance to pairwise losses (Saunshi et al. [27]).
> + **The loss functions listed in You et al. differ from our COLES**. Kindly see the analysis in the Resp. 2 to Rev. 1 (CRCP). **Our negative sampling realizes the logarithm of so-called geometric mean over RBF responses**, which is robust under low numbers of samples (our negative graph is very sparse). In contrast,  You et al. use the loss whose negative part can be seen as the arithmetic mean (as noted in Wang and Isola, ICML 2019).
> + For the node classification, without the use of  augmentations, **our S$^2$GC+COLES not only achieves better results but is 70-366x faster on CORA and Citeseer** compared to You et al. But we understand that work of You et al. is designed for the graph not node classification.
> + **Simplicity: S$^2$GC+COLES can be realized with few lines of code** ( command-line MATLAB), and they enjoy the closed-form solution without the need for GPU, complicated back-propagation rules, tuning of many parameters.
>
>
> ## 5. Tables 2 and 7 not solid enough to verify the effectiveness (lack of some state-of-the-art methods for comparison).
>
> + **We have listed results for all key contrastive methods** in our paper, e.g. DeepWalk, GCN+SampledNCE, SAGE+SampledNCE, Graph2Gauss, DGI, GRACE.
>
> + **We have now added results of You et al.** and also found one more work entitled `*Graph Contrastive Learning with Adaptive Augmentation*' by Y. Zhu et al. (WWW 2021), which we clearly outperform (**GCA method** listed already in the table above).
>
>
> + We believe the most meaningful comparison  is between for instance between GraphSAGE     vs. GraphSAGE+COLES, GCN vs. GCN+COLES, SGC vs. SGC+COLES, S$^2$GC vs. S$^2$GC+COLES etc. because then **the same backbone is ensured**, what only differs is the classifier type (we have provided that in the main paper).
>
> + We would be grateful if the esteemed reviewer could point to us which other methods that enjoy a similar problem and experimental setting to ours can be added into tables, and we are very happy to list them, but we struggled to find any other works which employ the  problem/experimental  setting as ours.
>
> **We hope not to be rejected just because we might have missed some recent paper** (that we are not aware of) from our tables. Our contributions are theoretical and universal, applicable and complementary to existing networks.

---

> ### Comment · Area_Chair_XLe1 · 2021-08-25
> **Reviewer response required**
>
> Dear Reviewer Gh8g,
>
> The authors have provided a very thorough response to your review. Has any of this changed your mind? Would you be willing to raise your score in response to their comments?

---

### Official Review · Reviewer_CRCP · 2021-07-17

**Rating:** 6
**Confidence:** 5

**Summary:**

The paper proposes a reformulation of Laplacian Eigenmaps in a log-linear contrastive setting. Inspired by GAN architecture, the authors show that the their contrastive laplacian eigenmaps (COLES) minimises a surrogate of Wasserstein distance as opposed to JS distance in traditional contrastive methods. Further, COLES in shown to enjoy block contrastive loss, which is upper bounded by pair-wise loss.

The authors show that the log-linear contrastive loss can be reformulated as a generalised eigenvalue (GEV) problem of a contrastive degree normalised adjacency matrix. They apply this formulation to line graph networks such as SGC and $S^{2}$GC and obtain a closed form solution to the matrix of node embeddings, which depends on a spectral filter and the solution to the GEV problem.

By casting the positive and negative samples as the real and generated data from a GAN, the authors show that traditional contrastive losses can be reformulated into JS divergence, while COLES can be cast into a surrogate of 1-Wasserstein  metric. They claim that the latter is preferable since, under poor overlap of the the two distributions JS divergence vanishes, leading to vanishing gradients, while 1-Wasserstein distance does not.

The authors also show that COLES is Lipschitz continuous and enjoys block contrastive loss, wherein one can leverage blocks of similar data as opposed individual pairs. Since from [Saunshi et al] block contrastive loss is smaller than pair-wise loss, the authors claim that COLES is expected to achieve a lower minimum.

The authors perform empirical evaluation on a host of open source datasets and multiple baseline backbones. In most tasks, COLES objective seems to outperform the methods included in the comparison.

**Limitations And Societal Impact:**

 Not applicable.

**Main Review:**

Originality:
The approach seems simple, straightforward but original. They use the log-linear contrast function, instead of the sigmoid, which allows reformulation as a generalised eigenvalue problem. If the contrastive learning problem is interpreted as a GAN, then convergence is in the Wasserstein topology which does not suffering from the vanishing gradient problem when the two measures do not have overlapping support.

Quality: The submission is technically sound, with extensive empirical evaluation and sufficient theoretical justification.  However, the authors do not sufficiently discuss the weakness of their approach. For example, the reason for the choice of log linear loss, or its interpretation. Is then Laplacian eigen map just one example of contrastive learning or are there some optimality results in this direction?

Clarity: The paper starts off clearly written but in some sections, feels disjointed. For example section 4.1 lines 128-152 could have been summarised succinctly as most of the derivation there is well known. The empirical evaluation is described adequately.

Significance:
The paper presents a thoughtful approach to contrastive graph embedding as generalised eigen value problem. If explored further, the results may provide better understanding of contrastive graph embedding  approach.


**Time Spent Reviewing:**

4

---

> ### Author Response · Authors · 2021-08-10
> **Response to Rev. 1 (CRCP)**
>
>
>
> We thank the reviewer for the constructive review and interesting questions regarding the nature of our loss.
>
> ## 1. The reason for the choice of log linear loss.
>
>
>
> + By analysing SampledNCE framework, **we choose the $\log(\exp(\cdot))$ because it lets us cast Eq. 3 as a simple quadratic problem** in Eq. 5 and 6, which enjoys a solution via the SVD decomposition (to deal with the orthogonality constraints).
> Thus, for COLES with linear networks (SGC and S$^2$GC), **we have a simple constrained quadratic problem with a unique solution** (no local minima) or vanishing gradients.
>
> + Quadratic optimization has been studied in many works and fast solvers based on Preconditioned Riemannian Optimization exist, e.g. `*Accelerated Optimization with Orthogonality Constraints*' (J. W. Siegel).
>
> + Under the chosen loss, **we can prove the  Lipschitz continuity of entire COLES objective, which casts it as the Kantorovich-Rubinstein duality for the Wasserstein distance**. **This is not generally true for the contrastive loss in Eq. 2 (generic SampledNCE)**.
>
>
> ## 2. Interpretation of the proposed loss.
>
> Below we provide the geometric interpretation of our loss with the following analysis:
>
> + It was shown in `*Distributed Representations of Words and Phrases and their Compositionality*' by Mikolov et al. that SampledNCE with the sigmoid non-linearity (their Eq. 4) is a practical approximation of SoftMax contrastive loss (their Eq. 2), the latter suffering poor scalability w.r.t. the count of negative samples. For this reason,  the majority of contrastive GNNs (DeepWalk, GraphSAGE, DGI, Graph2Gauss in Table 2, main paper) adopt SampledNCE rather than SoftMax (GRACE in Table 2, main paper).
>
> + Importantly, the **SoftMax contrastive loss has been decomposed** in `*Understanding Contrastive Representation Learning through
> Alignment and Uniformity on the Hypersphere*' by Wang and  Isola **into so-called $\mathcal{L}\_{\text{align}}$ and
>  $\mathcal{L}\_{\text{uniform}}$**:
> $$\mathcal{L}=-\log \frac{e^{\mathbf{u}^{\top} \mathbf{v} }}{e^{\mathbf{u}^{\top} \mathbf{v} }+\sum\_{u'\in\mathcal{N}} e^{\mathbf{u'}^{\top} \mathbf{v} }} = \mathcal{L}\_{\text{align}} + \mathcal{L}\_{\text{uniform}},(1)$$
> where $\mathcal{N}$ is a sampled subset of negative samples, $u$ and $v$ are node indexes of so-called positive sample and anchor embeddings $\mathbf{u}$ and $\mathbf{v}$, and we let $\langle\mathbf{u},\mathbf{u}\rangle=\langle\mathbf{u}',\mathbf{u}'\rangle=\langle\mathbf{v},\mathbf{v}\rangle=\tau^2$ ($\tau$ may act as the so-called temperature). **Of course, the total loss requires drawing $v$ and $u$ from the graph according to their connectivity and summing over multiple $\mathcal{L}$ but we skip this step for brevity**.
> $\mathcal{L}\_{\text{align}}=-\langle\mathbf{u}, \mathbf{v}\rangle$
> and $\mathcal{L}\_{\text{uniform}}=\log \sum\_{u^\ddagger\in\mathcal{N}\cup\\{u\\}} e^{\mathbf{u^\ddagger}^{\top}\mathbf{v}}$, that is, $\mathcal{L}\_{\text{uniform}}$ is a logarithm of an **arithmetic mean of RBF responses** over the subset $\mathcal{N}\cup\\{u\\}$.
>
> + In contrast, **COLES can be formulated as**:
> $$-\log  e^{\mathbf{u}^{\top} \mathbf{v} }-\frac{1}{|\mathcal{N}|}\sum\_{u'\in\mathcal{N}} \log e^{-\mathbf{u'}^{\top} \mathbf{v} } = %-\log e^{\mathbf{u}^{\top} \mathbf{v} }+\frac{1}{|\mathcal{N}|}\sum\_{u'\in\mathcal{N}} \log e^{\mathbf{u'}^{\top} \mathbf{v} } =
> \mathcal{L}\_{\text{align}} + \mathcal{L}'\_{\text{uniform}}=-\log\frac{e^{\mathbf{u}^{\top} \mathbf{v}}}{\left(\Pi\_{u'\in\mathcal{N}}e^{\mathbf{u'}^{\top} \mathbf{v}}\right)^{\frac{1}{|\mathcal{N}|}}},
> (2)$$
> where  $\mathcal{L}_{\text{align}}$ remains the same with SoftMax but $\mathcal{L}'\_{\text{uniform}}=\log\left(\Pi\_{u'\in\mathcal{N}}e^{\mathbf{u'}^{\top} \mathbf{v}}\right)^{\frac{1}{|\mathcal{N}|}}$ is in fact a **logarithm of the geometric mean  of RBF responses** over the subset $\mathcal{N}$. Thus, **our loss can be seen as the ratio of geometric means over RBF functions**.
>
> + Several authors have observed that **the geometric mean helps smooth out the Gaussian noise under the i.i.d. uniform sampling while loosing less information than the arithmetic mean**, e.g. see `Digital Image Processing 3nd Edition. Prentice Hall' by R. Gonzalez. This aligns with other  observations that **the geometric mean enjoys better confidence intervals for a small number of samples compared to the arithmetic mean**.
> + Given that **we sample few negative nodes for efficacy, we expect the geometric mean is more reliable**. Moreover, during random sampling, we cannot guarantee that the anchor class and classes of sampled negative nodes are  different (the very nature of contrastive learning). Thus, **if anchor and negative sample are of the same class, it is in fact an outlier (a noise) which the geometric mean is better equipped to deal with**.
>
>
>
> ## 3. Is the Laplacian Eigenmap just one example of contrastive learning or are there some optimality results in this direction?
>
> + In fact, **Eq. (1) and (2)** (in the above response) **are just two examples of a generalized loss** that we propose in this rebuttal and plan to study (e.g., in a journal extension):
> $$\mathcal{L}\_{\text{align}} + \mathcal{L}''\_{\text{uniform}}=-\log\frac{e^{\mathbf{u}^{\top} \mathbf{v}}}{M\_p\left(
> e^{\mathbf{u'\_1}^{\top} \mathbf{v}},\cdots,e^{\mathbf{u'^{\top}\_{|\mathcal{N}|}} \mathbf{v}}\right)},$$
> where $M_p(\cdot)$ is the so-called generalized mean, e.g. if $p=0$, one has the geometric mean, if $p=1$, one has the arithmetic mean, if $p=-1$, one has the harmonic mean, etc. However, **only for the geometric mean, Contrastive Laplacian Eigenmaps emerge**. The **geometric mean has better confidence intervals than arithmetic mean for low sample size** (our negative graph is sparse).
> \
> \
> To make the discussion interesting, we have evaluated several variants (different $p$) for the above equation, and we have the following table which strongly supports that the geometric mean emerging in COLES is very robust comared to other types of mean:
> |                           | Cora (5)  | Cora (20) | Citeseer (5) | Citeseer (20) |  Pubmed (5)  | Pubmed (20) | Cora-full (5) | Cora-full (20) |
> |---------------------------|-----------|-----------|--------------|---------------|-----------|-----------|--------------|---------------|
> | **Geometric ($M_0$  + S$^2$GC (COLES))**                | 76.5±2.6  | 81.5±1.2  | 67.5±2.2     | 71.3±1.0      | 66.0±5.2  | 77.4±1.9  | 50.8±1.4     | 61.8±0.5      |
> | Arithmetic ($M_1$ + S$^2$GC (Softmax-Contrastive))                      | 71.8±3.0   | 77.6±1.3  | 63.2±3.1     | 69.3±0.8     | 65.9±4.3   | 77.1±1.5  | 49.2±1.4     | 60.6±0.6     |
> | Harmonic ($M_{-1}$ + S$^2$GC)                     | 75.2±3.5   | 80.7±1.2  | 64.7±2.4     | 70.9.±0.9     | 65.9±5.5   |  73.9±2.4   |  48.0±1.6      |  59.7±1.6      |
> | Quadratic ($M_{2}$ + S$^2$GC)                     | 72.3±2.5   | 77.2±1.3  | 65.4±2.2     | 70.7.±0.8     | 65.6±4.5   |  77.3±1.5   |  49.2±1.5      |  60.6±1.6      |
>
>
> + Moreover, **we notice that standard SoftMax contrastive learning**, like with any SoftMax formulation, it **suffers from regions of flatness** (think the sigmoid function), **which will result in the vanishing gradients** in some neighbourhoods.
> Note that different methods typically only differ in the $\mathcal{L}\_{\mathrm{uniform}}$
> term. By simple derivations,
> $$d\mathcal{L}\_{\mathrm{uniform}}= \frac{e^{\mathbf{u}^\top\mathbf{v}}}{e^{\mathbf{u}^\top\mathbf{v}} + \sum\_{u'}e^{\mathbf{u}'^\top\mathbf{v}}}d{\mathbf{u}^\top\mathbf{v}}+\sum_{u'}\frac{e^{\mathbf{u}'^\top\mathbf{v}}}{e^{\mathbf{u}^\top\mathbf{v}} + \sum_{u'}e^{\mathbf{u}'^\top\mathbf{v}}}d{\mathbf{u}'^\top\mathbf{v}}
> =p\_{\mathbf{u}} d{\mathbf{u}^\top\mathbf{v}}+ \sum_{u'} p\_{\mathbf{u}'} d{\mathbf{u}'^\top\mathbf{v}},$$
> which gives the backpropagated gradient of ${\mathbf{u}^\top\mathbf{v}}$.
> In such regions where $e^{\mathbf{u}'^\top\mathbf{v}}\ll{e}^{\mathbf{u}^\top\mathbf{v}}$,
> the coefficients $p_{\mathbf{u}'}$ in the second term will vanish, meaning
> the gradient associated with the negative samples is blocked.
> In contrast, **COLES is quadratic and it enjoys a unique optimum**.
> We have:
> $$d\mathcal{L}'\_{\mathrm{uniform}}=\sum\_{u'}{} \frac{1}{\vert\mathcal{N}\vert} d(\mathbf{u}'\mathbf{v}).$$
> The backpropagation follows a simple rule with constant coefficients.
>
> + Kindly also see Resp. 5 to Rev. 4 (8s4v) (e.g., **how we explore the subset of functions satisfying the Lipschitz condition**).
>
>
> ## 4. Summarise lines 128-152 succinctly.
>
> Absolutely. As per comment, we have now revised that part of text by removing Eq. 8 and 9 (prior knowledge) and compressing the surrounding text. We just wanted to politely highlight that **the idea of analysing SampledNCE by plugging it into the WGAN framework is nonetheless our contribution** explaining why COLES performs well.
>
>
> **We hope that the above several perspectives on the Contrastive Laplacian Eigenmaps, including the geometric interpretation, are convincing and enjoyable** food for thought.

---

### Author Response · Authors · 2021-08-21
**Rolling Discussions: How can we help?**

Dear Reviewers,
\
Esteemed Area Chair,

We would like to let you know that **we are here ready (and more than happy) to answer any further questions that you may have** regarding our work in the spirit of **the rolling discussions**, as indicated by Program Chairs in one of their e-mails:

'to minimize the chance of misunderstandings during the reviewing process, there will be a rolling discussion after the initial response period and you will be able to respond to any reviewer questions that arise during the discussion'.

Kind regards,
\
Authors

---

### Decision · Program_Chairs · 2021-09-27

**Decision:**

Accept (Poster)

**Comment:**

This paper had very split reviews and required some thought when coming to a recommendation. The authors provided very thorough responses to each reviewer's critiques. While the two positive reviews responded saying they were satisfied with the author responses, neither of the negative reviewers responded to the author feedback. One negative reviewer's only objection was a lack of comparison against You et al. 2020. The author's provided such a comparison in the response, showing good results for their method relative to the baseline. The other reviewer had more detailed critiques, but from my estimation of the author's response, those critiques were well rebutted. Unfortunately I do not know if the reviewer themself thought the critiques were answered well. I am inclined to recommend acceptance for this paper, on the basis of the very thorough responses which I believe answered most of the reviewer critiques, and the failure of the reviewers themselves to answer the authors.